# Global changes of the RNA-bound proteome during the maternal-to-zygotic transition in *Drosophila*

Vasiliy O. Sysoev[1], Bernd Fischer[1,2], Christian K. Frese[1], Ishaan Gupta[1], Jeroen Krijgsveld[1], Matthias W. Hentze[1], Alfredo Castello[1,3] & Anne Ephrussi[1]

The maternal-to-zygotic transition (MZT) is a process that occurs in animal embryos at the earliest developmental stages, during which maternally deposited mRNAs and other molecules are degraded and replaced by products of the zygotic genome. The zygotic genome is not activated immediately upon fertilization, and in the pre-MZT embryo post-transcriptional control by RNA-binding proteins (RBPs) orchestrates the first steps of development. To identify relevant *Drosophila* RBPs organism-wide, we refined the RNA interactome capture method for comparative analysis of the pre- and post-MZT embryos. We determine 523 proteins as high-confidence RBPs, half of which were not previously reported to bind RNA. Comparison of the RNA interactomes of pre- and post-MZT embryos reveals high dynamicity of the RNA-bound proteome during early development, and suggests active regulation of RNA binding of some RBPs. This resource provides unprecedented insight into the system of RBPs that govern the earliest steps of *Drosophila* development.

[1] European Molecular Biology Laboratory, Meyerhofstrasse 1, 69117 Heidelberg, Germany. [2] German Cancer Research Center, Im Neuenheimer Feld 580, 69120 Heidelberg, Germany. [3] Department of Biochemistry, University of Oxford, South Parks Road, Oxford OX1 3QU, England. Correspondence and requests for materials should be addressed to A.E. (email: ephrussi@embl.de).

The maternal-to-zygotic transition (MZT) occurs in animal embryos during the first stages of development. In this process maternally deposited molecules are degraded and gradually replaced by products of the zygotic genome[1–3]. However, zygotic genome activation does not occur immediately upon fertilization: for example, in *Drosophila*, the zygote's DNA remains mostly transcriptionally silent for hours after pronuclear fusion, and early development is controlled by maternal messenger RNAs (mRNAs) and the RNA-binding proteins (RBPs) that regulate them. RBPs constitute a heterogeneous class of proteins that control all aspects of an mRNA's lifecycle. A dynamic set of RBPs associates with and accompanies nascent transcripts throughout maturation, localization, translation and decay. Previous attempts to create a census of RBPs of a cell relied on homology searches and *in vitro* approaches. While computational analyses contributed to annotate the compendium of RBPs harbouring classical RNA-binding domains, their capacity to discover unorthodox RBPs was limited[4]. *In vitro* protein–RNA screens reported RNA binding not only of canonical RBPs, but also of proteins previously unrelated to RNA biology[5,6] However, the promiscuity of protein–RNA interactions under *in vitro* conditions calls for validation of these unconventional RBPs in a physiological environment.

Recently, a proteome-wide method for RBP identification was developed and applied successfully to different cell lines[7–9], budding yeast[10,11] and the nematode *C. elegans*[11]. This method, named 'RNA interactome capture'[12] solves some of the limitations described above, by isolation and identification of RBPs in living cells. The method involves ultraviolet cross-linking of proteins to nucleic acids in live cells, followed by the capture of polyadenylated (poly(A)+) RNAs via hybridization to oligo(dT) beads under denaturing conditions. The proteins directly bound to poly(A)+ RNAs are then identified and quantified by mass spectrometry (MS).

We have further developed RNA interactome capture to enable identification of the RBPs functioning in a living multicellular organism, the *Drosophila* embryo. Beyond providing a comprehensive compendium of *Drosophila melanogaster* RBPs, we have applied this method to embryos at two different developmental stages employing a multiplexed proteomic approach, generating, to our knowledge, the first dynamic RNA interactome until now. Comparison of samples prepared from early and late embryos reveals that the RNA interactome undergoes important changes during development. This plasticity of the RNA-bound proteome can be explained either by changes in overall protein abundance or modulation of the RNA-binding activity of RBPs. Our analysis provides new insight into the molecular features of early development by highlighting proteins that may link development to RNA metabolism.

## Results

### Capture of RBPs in the *Drosophila* embryo.
To explore the landscape of RBPs in the early *Drosophila* embryo we further developed the RNA interactome capture method established by Castello *et al.*[8,12], that was initially optimized for cell lines. This method employs *in vivo* cross-linking of protein–RNA complexes and subsequent capture on oligo(dT) magnetic beads. Irradiation with 254 nm ultraviolet light activates the nucleotide bases that then rapidly form stable covalent bonds with some amino acid residues, preferentially nucleophilic and aromatic, placed at 'zero distance'[13], and does not induce detectable formation of protein–protein cross-links. Although 254 nm ultraviolet light does not reach the deepest volume of the embryo[14], it was the only agent suitable for reproducible and efficient cross-linking of RNA to protein *in vivo* available to us. Capture of poly(A)+

RNA by hybridization with oligo(dT) probes is compatible with the use of high salt concentrations and ionic detergents for efficient removal of non-covalently bound proteins. The resulting samples are compatible with tandem mass tag (TMT) labelling for multiplexed quantitative proteomics using Orbitrap MS combined with rigorous statistical methods to catalogue proteins as high-confidence RBPs (Fig. 1a).

After testing a spectrum of cross-linking conditions, we selected 1 J cm$^{-2}$ of 254 nm ultraviolet light as the lowest dose of ultraviolet light resulting in sufficient protein–RNA cross-linking (Supplementary Fig. 1a–f, discussed in the Supplementary Note 1) without causing detectable RNA damage (Fig. 1b). Also, ultraviolet doses up to 1.5 J cm$^{-2}$ did not result in tubulin contamination. Furthermore, quantitative analysis by Agilent Bioanalyzer 2100 (Fig. 1b) showed that the captured polyadenylated (polyA+) RNA was depleted of highly abundant non-coding RNAs: transfer RNAs were undetectable and ribosomal RNA (rRNA) amounts reduced to <10% of the captured RNA, when compared to whole-embryo total RNA (Fig. 1b). Reverse transcription quantitative polymerase chain reaction (RT-qPCR) confirmed a multifold enrichment of both the abundant *gapdh* mRNA and the low abundance *ts* mRNA over 18S rRNA (Fig. 1c).

Gel electrophoresis combined with silver staining revealed that oligo(dT)-bound fractions recovered from ultraviolet-irradiated, cross-linked (CL) embryos contained a complex protein mixture distinct from the total lysate, whereas the non-cross-linked (noCL) controls were essentially devoid of proteins (Fig. 1d). Specific purification of well-known RBPs was confirmed by western blotting (Fig. 1e). Vasa, eIF4E, Pabp2, PABPC and Hrp48 were strongly enriched in eluates only when ultraviolet light was applied to the embryos, while typical contaminants such as histone H3 and RNP components that do not directly bind RNA such as exon junction complex component Y14 (ref. 15) and kinesin[16] were depleted in our samples. However, in some cases we could detect a faint α-tubulin band in ultraviolet-irradiated samples (red asterisk, Supplementary Fig. 1h). We further improved the stringency of lysis by increasing DTT concentration to 12.5 mM and incubating lysates at 60 °C for 15 min before oligo(dT) bead addition, which resulted in complete removal of α-tubulin from ultraviolet-irradiated samples. This additional treatment did not induce detectable protein (Supplementary Fig. 1i–p) or RNA (Supplementary Fig. 1g) degradation, suggesting that high temperature enhances protein denaturation and improves removal of non-covalent RNA binders.

### Determination of the *Drosophila* RNA interactome.
To obtain a comprehensive view of RBPs in *Drosophila*, we applied RNA interactome capture to early, pre-MZT (0–1 h post-fertilization) and late (4.5–5.5 h), post-MZT embryos (Fig. 1f). By visual inspection using DAPI-staining we confirmed that more than 95% of embryos were at the expected stage (Supplementary Fig. 1q–s). For total interactome determination we pooled early and late samples to obtain three biological replicates—three CL and three noCL, control samples—that were analyzed in parallel using TMT MS with six different isobaric labels (TMT MS3 (ref. 17), Methods and Supplementary Note 2). This resulted in identification of 2,336 peptides, 2,200 of which could be mapped to a unique protein model, leading to 978 proteins identified. Protein intensity ratios in CL over noCL samples were calculated for 187 proteins, 65 of which were significantly enriched (1% false discovery rate (FDR), red dots Fig. 1g; Supplementary Fig. 1t,u). Quantification of the remaining proteins was impossible by the mentioned method due to extremely low abundance of these

proteins in the control (noCL) samples, which resulted in the absence of intensity values required to calculate protein abundance ratios.

To overcome the technical difficulty of missing intensity values in the noCL samples, we performed a semi-quantitative analysis taking into consideration the number of peptide occurrences in ultraviolet-irradiated and non-irradiated samples for each protein across the three biological replicates (Supplementary Fig. 1v). The

number of peptide occurrences in the negative controls and ultraviolet-irradiated samples was used to compute the incidence of false-positives and to estimate the FDR. By this analysis only peptides occurring predominantly and reliably (FDR 1%) in CL samples over the noCL controls were considered positive hits (Supplementary Fig. 1v). In total 523 proteins qualified as high-confidence RBPs by direct comparison of protein ratios or semi-quantitative analysis (Fig. 1h). Hereafter, we refer to this set

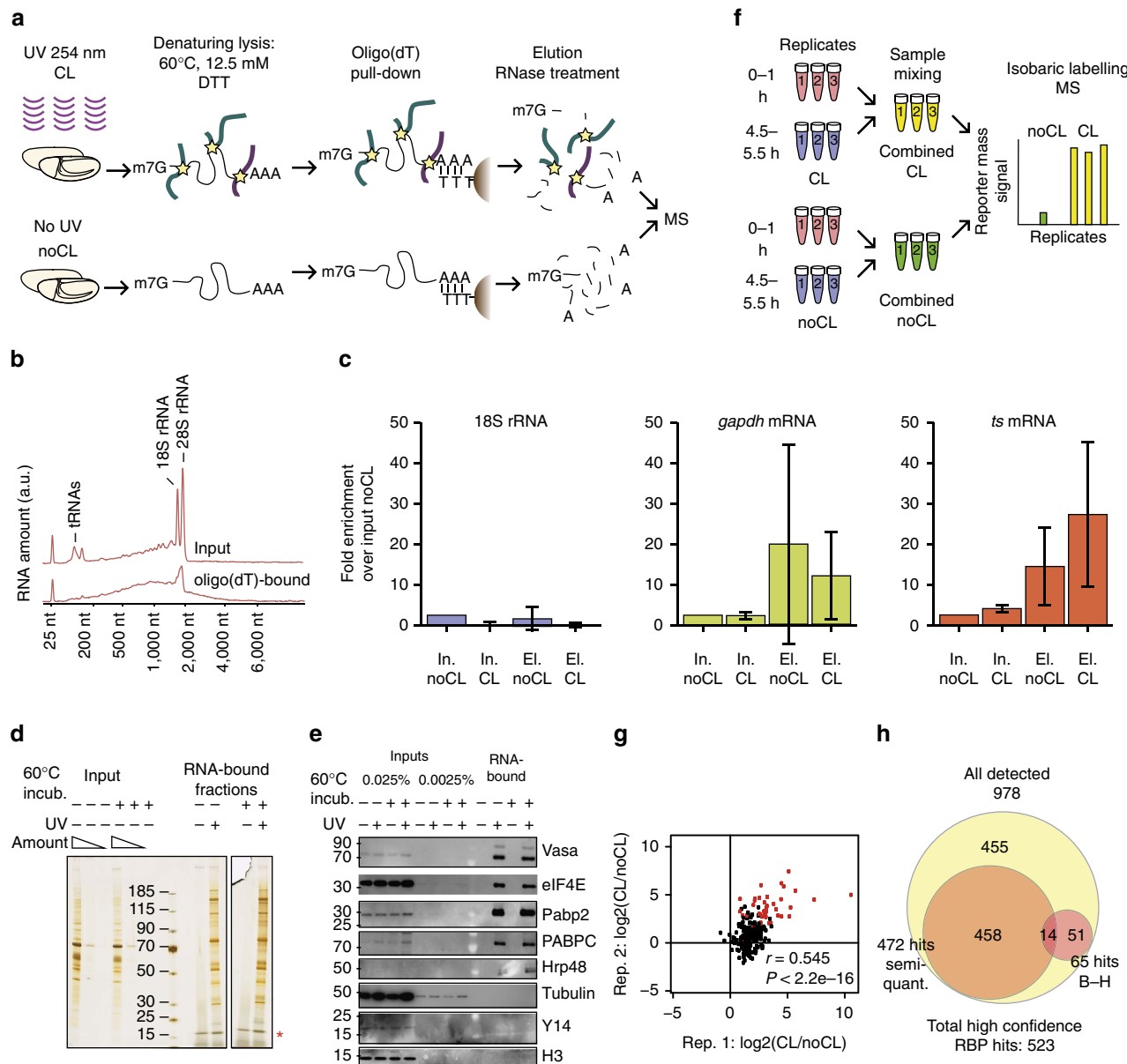

**Figure 1 | Identification of the *Drosophila* RNA interactome.** (**a**) Direct RNA binders are CL to mRNAs in living *Drosophila* embryos, which are subsequently lysed under denaturing conditions. mRNA-protein complexes are purified by hybridization with oligo(dT) magnetic beads and a series of stringent washes. Proteins are released by RNase treatment and are ready for MS analysis. (**b**) Analysis of total and oligo(dT) by Bioanalyzer 2100 captured RNA shows depletion of abundant ncRNAs. (**c**) Multifold enrichment of polyadenylated *gapdh* and *ts* mRNAs in oligo(dT) bound fractions confirmed by RT-qPCR. On the *x* axis are indicated the samples in which the amounts of the different RNAs were measured: Input noCL, input CL, eluate noCL, eluate CL. The *y* axis represents the fold enrichment of RNA amounts in the different samples. RNA amounts in noCL input were defined as 1. Error bars: s.d. See Supplementary Table 1 for information on oligonucleotides used for the analysis. (**d**) Protein profiles of total embryo and RNA-bound fractions. (**e**) Analysis of total embryo lysates and oligo(dT) bound fractions by western blotting with antibodies against Vasa, eIF4E, Pabp2, PABPC, Hrp48, tubulin, Y14 and H3. Lysis at 60 °C and 12.5 mM DTT ensures that tubulin background is removed. See Supplementary Table 2 for antibody information. (**f**) Replicated samples prepared from pre- (0–1 h) and post-MZT (4.5–5.5 h) embryos are mixed to generate three aggregate CL and noCL (control) samples. Proteins are partially digested, TMT labelled and quantified by MS. (**g**) Scatter plot showing protein abundance ratios CL/noCL in two replicates. Red dots represent proteins significantly enriched in CL samples. (**h**) Venn diagram comparing numbers of detected and significantly enriched proteins.

of proteins as the *Drosophila* mRNA interactome (Supplementary Data 1).

**Discovery of hundreds of novel *Drosophila* RBPs.** As expected, gene ontology (GO) term enrichment analysis of the RNA interactome, as compared to the total lysate, revealed enrichment of biological processes in which RNA has a key role, such as translation and RNP structure organization (Supplementary Fig. 2a). We counted the RNA interactome proteins annotated by the GO term 'RNA binding' as well as other RNA-related GO terms (any GO term that contains 'RNA', for example, 'RNA splicing') (Fig. 2a). RNA interactome capture 're-discovered' 136 proteins (of 523; 26%) previously annotated as binding RNA, while in the total proteome the number of such proteins was 293 (of 3,126 proteins identified in the total lysate, not including interactome proteins; 9%; for the analysis of total proteome see below). A total of 134 (out of 523; 26%) RNA interactome proteins were not annotated by the GO term 'RNA binding' but were nevertheless annotated with other RNA-related GO terms (Fig. 2a). The number of proteins falling into this category in the

total embryo proteome was 639 (of 3,126; 20%). Interestingly, the remaining 253 RNA interactome proteins could not be linked to RNA biology based on current GO annotation. Although 17 of these do harbour a well-established RNA-binding domain (RBD) according to the Pfam database, the other 236 lack known RBDs and are hereafter referred to as novel or newly discovered RBPs (Fig. 2b).

A parallel study reports the RNA interactome of early *Drosophila* embryos[18]. The total number of high-confidence RBPs identified by Wessels *et al.* (476) is comparable to the number we identified and also includes many proteins not previously shown to bind RNA (99).

**Low complexity and disordered regions in *Drosophila* RBPs.** The RNA interactome comprises 47 distinct types of known RBDs, as expected of a catalogue of *bona fide* RBPs. Compared to the total embryo lysate, our dataset is significantly enriched in canonical RBDs such as RNA recognition motif (RRM), ribosomal and K-homology (KH) domains (Supplementary Fig. 2b,c). Note that most previously known RBDs, such as the

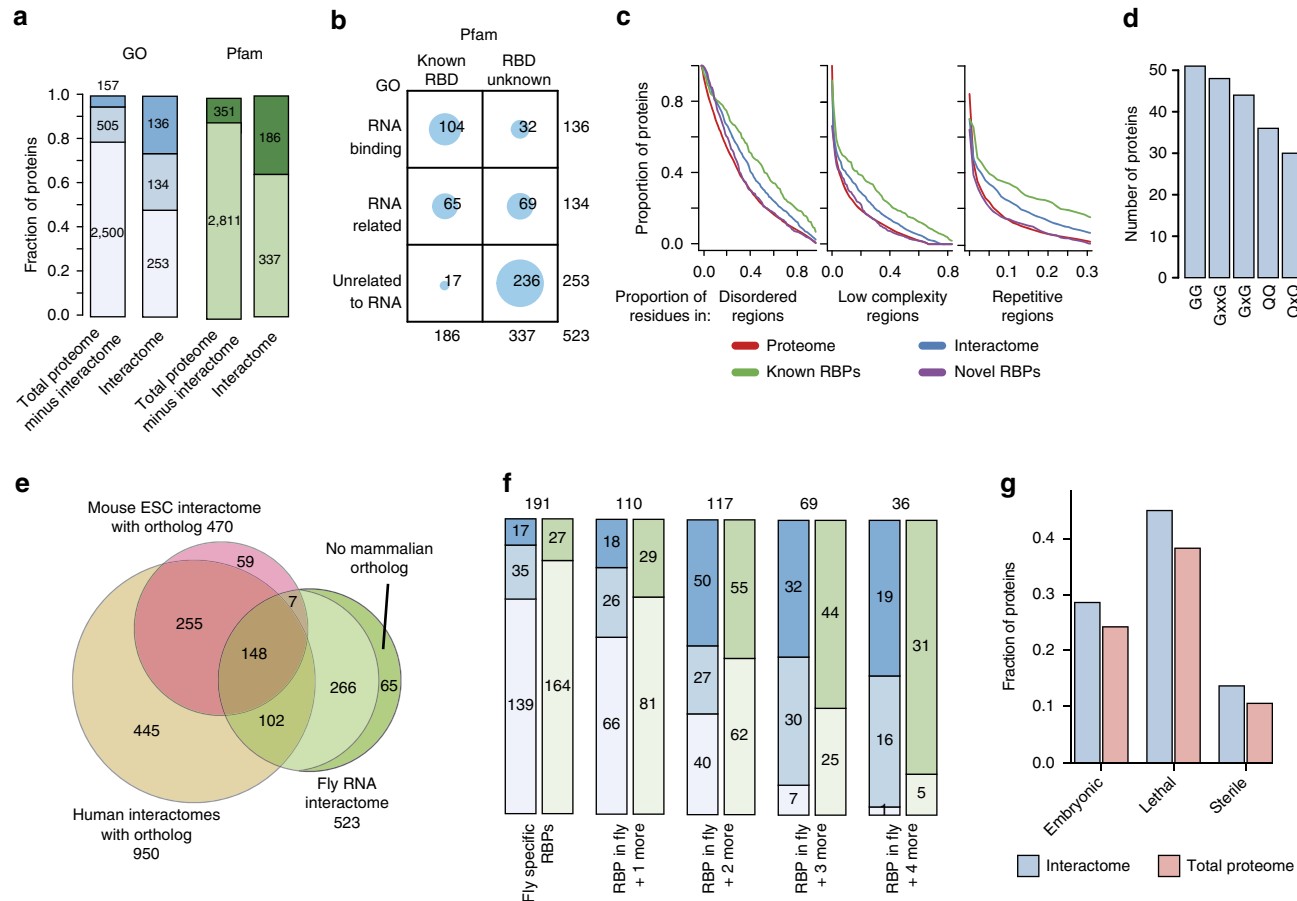

**Figure 2 | Composition of the *Drosophila* mRNA interactome and its links to development.** (**a**) Blue bars: numbers of proteins annotated with GO term 'RNA binding' and other RNA-related GO terms. Green bars: numbers of proteins containing and not containing domains annotated as RNA-binding in the Pfam database. Dark blue: GO term 'RNA binding'; medium blue: RNA-related GO terms; light blue: GO annotation unrelated to RNA; dark green: known RBD; light green: no known RBD. (**b**) Balloon plot related to **a**. (**c**) Distribution of disordered regions, low complexity domains and repetitive sequences in the total proteome (red), RNA interactome (blue), previously known RBPs (green) and newly discovered RBPs (purple) in the RNA interactome. Two-sample Kolmogorov–Smirnov test *P* values are listed in the Supplementary Note 3. (**d**) Five most frequent repetitive sequence patterns in the RNA interactome. (**e**) Venn diagram showing numbers of proteins belonging to the human (yellow), mouse (pink) and fly (green) interactomes. Of 266 fly-specific RBPs 65 do not have a mammalian ortholog. (**f**) Blue bars: fly-specific RBPs and RBPs shared with one or more eukaryotic interactomes are compared to RNA-related GO terms (blue bars). Green bars: numbers of proteins with Pfam-annotated known RBDs. (**g**) Fractions of proteins with embryonic, lethal and sterile phenotypes in the RNA interactome (blue) and the proteome (red). Fischer test *P* values resulting from comparison of the interactome to the proteome: lethal—0.00346; embryonic—0.03483; sterile—0.0001754.

La domain, dsRBP (DSRM) and zinc-finger domains, were overrepresented in the RNA interactome compared to the whole embryo lysate but their enrichment did not qualify as statistically significant due to the small number of proteins containing such domains in the *Drosophila* embryo proteome (Supplementary Fig. 2b).

Of the 523 *Drosophila* RNA interactome proteins, 70 (of 287) previously known and 37 (of 236) newly discovered RBPs contain repetitive motifs in intrinsically disordered regions, which are protein segments lacking a stable three-dimensional structure, typically of low amino acid complexity (Fig. 2c,d; Supplementary Fig. 2d; Supplementary Note 3). Such regions were previously reported to be overrepresented among RBPs in mammals[8,9]. However, in contrast to mammals, the most prevalent repetitive regions in *Drosophila* RNA interactome are based on the neutral amino acids glycine, glutamine and asparagine (Fig. 2d; Supplementary Fig. 2d).

Although the whole RNA interactome is enriched in disordered, low complexity and repetitive regions compared to the total proteome, the novel proteins are on average less disordered than previously known RBPs (Fig. 2d). A similar pattern was observed for the RBPs identified in mouse embryonic stem cells[9], while novel human RBPs were, on the contrary, more disordered than those previously known[8]. These observations might reflect differences in RBP functions, the mechanisms that drove evolution of RBPs in these species, or annotation biases. Although our observations call for further investigation, we find it probable that the difference in disorder of novel and previously known *Drosophila* RBPs is not caused by experimental bias, as other parameters such as average length, isoelectric point and hydrophobicity are similar for these two groups (Supplementary Fig. 2e–g). In addition, amino acid composition of the novel and previously known RBPs is similar, and the significant enrichment of positively charged residues observed in both groups is characteristic of RBPs (Supplementary Fig. 2h,i).

***Drosophila* and other eukaryotic interactomes compared**. We compared the *Drosophila* RNA interactome to the compendium of human (sum of HeLa[8], HEK293 (ref. 7) and HuH7 (ref. 10) data sets), mouse[9], worm[11] and yeast (sum of refs 11 and 10) RNA interactomes. Supplementary Fig. 2j shows a Venn diagram comparing all five available RNA interactomes, and Fig. 2e a Venn diagram of mouse, human and fly data. A total of 336 out of 523 proteins were detected as RBPs in at least one of the 4 species and 105 proteins in at least 3 species (Fig. 2e; Supplementary Data 2). Comparison to the GO term 'RNA binding' and other RNA-related GO-terms (Fig. 2f, blue bars) revealed that these proteins are enriched in core elements of the splicing and translation machineries: for example, 32 structural ribosomal proteins constitute 22% of the common fly, human and mouse RBPs, while this class constitutes only 6% of fly-specific RBPs (12 of 191). In agreement with this observation, proteins harbouring classical RBDs were highly prevalent across the proteins shared by all four RNA interactomes—(86%, 31 of 36) while the number of proteins harbouring known RBDs is lower among fly embryo-specific RBPs (14%, 27 of 191).

**Novel *Drosophila* RBPs are involved in development**. We used published information to evaluate possible links between RBPs we identified and development. In our RNA interactome capture experiment, half the sample was generated from 4.5 to 5.5 h embryos (Fig. 1b) in which the first cell types differentiate[19], including neuroblasts[20]. We used mRNA localization data[21] to identify mRNAs encoding RBPs expressed in neuroblasts and that might be involved in neuroblast specification and nervous system

development. Among the 24 mRNAs localized to neuroblasts, three encode novel RBPs, including HmgD, Dpa and Mcm2. Mcm2 and 42 other novel RBPs discovered in this study were also identified in an RNAi screen[22] as having roles in neuroblast development.

Next, we took advantage of the wealth of phenotypic data for *D. melanogaster* in Flybase[23] and extracted phenotypic descriptions for alleles of genes encoding RNA interactome proteins. The numbers of proteins for which at least one allele is associated with an embryonic, lethal or sterile phenotype are shown in Fig. 2g: 150 (29%) are associated with embryonic phenotypes, 236 (45%)—with lethal phenotype and 72 (14%)—with sterility. Note that protein sets falling into each phenotypic class often overlap: for example, among 150 proteins with embryonic phenotypes 144 were also lethal (Supplementary Fig. 2k). We also calculated fractions of proteins with which phenotypes are associated in the RNA interactome and the total proteome and found that for each of the three considered phenotypic classes this fraction was higher in the interactome (Fig. 2g). We classified proteins in each of the three phenotypic classes according to their molecular function using GO annotation. Among the proteins with lethal phenotypes, 15 had no assigned molecular function (Supplementary Fig. 2l). Interestingly, based on their mutant phenotypes, 9 of these are implicated in nervous system development: Fax, CSN1b, Nedd8, Ars2, Nopp140, Nop56, CG6066, Tap42, Patr-1.

Discovery of several metabolic enzymes as RBPs in *Drosophila* is consistent with findings of other interactome capture studies[8,10,11] and is noteworthy, as energy production was shown to be an important factor in developmental processes in some organisms[24,25], and its regulation might have a role in the MZT in *Drosophila*[26] As for the mammalian and yeast interactomes[10,11], our *Drosophila* interactome contains a substantial number (47) of proteins listed as enzymes (http://enzyme.expasy.org/). Whereas in yeast all steps of the glycolytic pathway can be catalyzed by enzymes identified as RNA binders, some of them binding mRNAs encoding glycolytic enzymes[10,11], in *Drosophila* only two glycolytic enzymes—phosphofructokinase and phosphoglyceromutase—are found in the RNA interactome (Supplementary Data 3). Further experiments on individual proteins will elucidate the role(s) of RNA binding by metabolic enzymes in development.

**Cell cycle and cytoskeleton regulators CycB and EB1 are RBPs**. We confirmed the RNA-binding ability of two newly discovered RBPs using a reciprocal approach that involves ultraviolet cross-linking of RNA to proteins, purification of proteins of interest and detection of bound RNA[7,27,28]. As targets for validation we chose cyclin B (CycB) and end-binding protein 1 (EB1), two proteins with important roles in the early embryonic mitotic divisions—a process that temporally overlaps and is impacted on by the MZT[29,30].

We employed flies expressing transgenic EB1-GFP, endogenously tagged GFP-CycB or free GFP. After ultraviolet crosslinking of live embryos, each candidate protein was immunoprecipitated and purified under stringent conditions using the anti-GFP nanobody[31]. Upon immunoprecipitation and extensive washing, we obtained high yields of GFP proteins, as illustrated by the case of free GFP in Fig. 3a. Following partial digestion, CL RNA was $^{32}$P labelled using T4 polynucleotide kinase (PNK), and the protein–RNA complexes were visualized by immunoblotting.

Immunoprecipitates of free GFP from both CL and noCL embryos did not contain radioactively labelled proteins in the expected size range (Fig. 3b). Immunoprecipitates from ultraviolet-irradiated GFP-CycB and EB1-GFP samples yielded

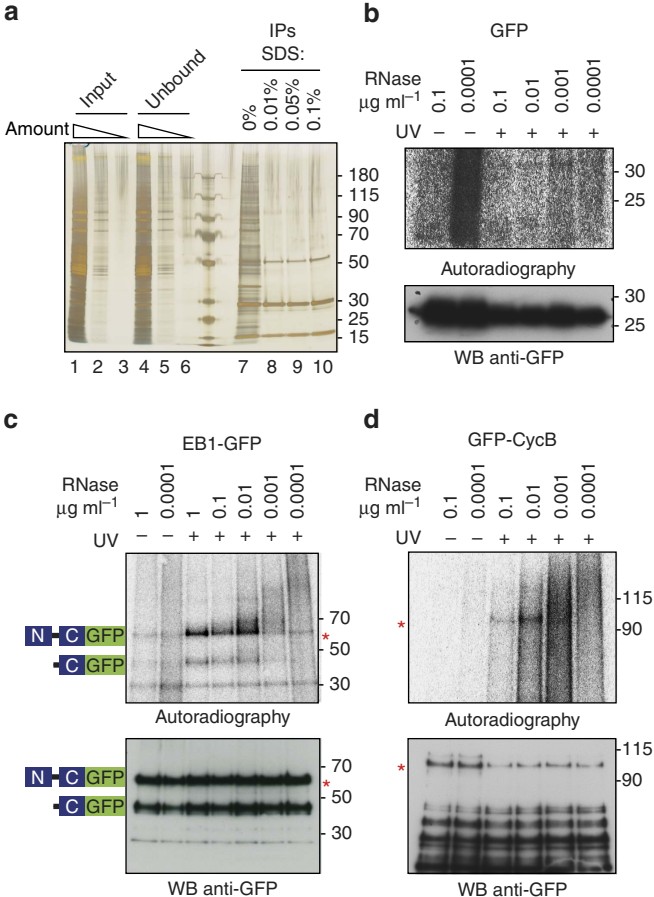

**Figure 3 | Validation of CycB and EB1 as RNA-binding proteins.**
(**a**) Optimization of GFP immunoprecipitation. Protein profiles of total embryo lysates, IPs of free GFP and unbound fractions are visualized on a silver stained polyacrylamide gel. Lanes 1–3: serial dilution of total embryo lysate expressing free GFP; lanes 4–6: serial dilution of unbound fractions after GFP-IP; lanes 7–10: GFP immunoprecipitates washed with buffers containing increasing concentrations (none; 0.01%; 0.05%; 0.1%) of SDS. (**b–d**) Lysates containing free GFP, EB1-GFP and GFP-CycB were treated with various concentrations of RNase A. GFP proteins were IPed from noCL and CL embryos, radioactively labelled, separated on polyacrylamide gels and blotted on PVDF membranes. Upper panels: autoradiography of membranes containing IPs of GFP (**b**), EB1-GFP (**c**) and GFP-CycB (**d**) labelled with $\gamma$-[$^{32}$P]-ATP by T4 polynucleotide kinase (see Methods section, PNK assay). Lower panels: visualization of GFP proteins by WB with an anti-GFP antibody.

discrete radioactive bands after PNK assay at the positions expected for each fusion protein (Fig. 3c,d), and the intensity of these bands was stronger in CL than in noCL samples, and the mass of radioactively labelled complexes depended on RNase A dose, as expected of *bona fide* RBPs. Immunoblotting confirmed the efficient immunoprecipitation of both GFP-CycB and EB1-GFP, as well as smaller products, presumably proteolytic fragments (Fig. 3c,d, lower panels). A presumed degradation product of EB1-GFP of ~45 kDa was radioactively labelled, along with the full-length protein, in a ultraviolet-dependent, RNase-dependent manner, which is indicative of its RNA-binding activity. Because this fragment is fused to GFP, it should include the C-terminal APC-binding domain of EB1-GFP[32].

**Dynamics of the RNA-bound proteome during the MZT.** To evaluate the changes in composition of the RNA-bound proteome

that take place during the MZT, we compared 0–1 h to 4.5–5.5 h ultraviolet-irradiated embryos (Fig. 4a). Protein profiles of RNA-bound fractions at the two stages were different, as revealed by PAGE and silver staining (Fig. 4b). Three independent biological replicates prepared from embryos collected at these two time windows were analyzed in parallel by TMT MS3 (see Methods section) using six different isobaric labels (Fig. 4c). Unique peptides were assigned to a total of 1,131 proteins, which included the majority of previously identified RNA interactome proteins (490 of 523). We calculated protein intensity ratios in late (4.5–5.5 h) versus early (0–1 h) RNA-bound fractions for all proteins detected (994) and identified 116 proteins that were significantly enriched at one of the stages with 10% FDR and 12 with 1% FDR, (Fig. 4d; Supplementary Fig. 3a,b).

To test whether such enrichment in the RNA-bound fraction merely reflects a change in proteins' overall abundance during development, or whether their RNA-binding activity is modulated, we analyzed whole proteome of 0–1 h and 4.5–5.5 h embryos, following a similar MS approach (Fig. 4c), and calculated late/early protein intensity ratios. A total of 1,033 proteins (28% of 3,655 proteins constituting the total proteome) were significantly enriched at one of the developmental stages: 785 with 10% FDR and 248 with 1% FDR (Fig. 4e; Supplementary Fig. 3c,d). We plotted the early/late abundance ratio of each protein in the RNA-associated fraction (ordinate) versus its early/late abundance ratio in the whole embryo proteome (abscissa) (Fig. 4f). Three classes of proteins emerged from this analysis: class 1: proteins whose abundance in the RNA-bound fraction and in the total proteome remain unaltered (grey dots; 1,015 proteins of which 405 are in the RNA interactome); class 2: proteins whose change in abundance in the RNA-bound fraction correlates with the abundance change in the whole proteome (black dots; 78 proteins, 31 of which are in the interactome); and class 3: proteins whose abundance in the RNA-bound fraction changed although their overall abundance in the whole embryo remains unaltered. This class also includes 6 proteins whose abundance in the RNA-bound fraction was reduced at the later stage, but their overall abundance in the embryo increased during development. Class 3 totalled 38 proteins, 27 of which are in the RNA interactome. We refer to these class 3 proteins as 'dynamic binders.' (Fig. 4g). Interestingly, classes 2 and 3 were enriched in previously known RBPs as compared to class 1, which contains the majority of the novel RBPs (Supplementary Fig. 3e). The majority of differentially expressed RBPs (50 of 78 class 2 proteins) and 35 of the 38 'dynamic binders' were enriched in the RNA-bound fraction of early, pre-MZT embryos. Results of our proteomic analysis are presented in Supplementary Data 4.

Interestingly, the single GO term enriched among the 38 dynamically regulated RBPs compared to all proteins identified in the RNA-bound fraction is 'mRNA splicing' (Fig. 5a). Eight of these are known splicing factors (Snf, Prp8, X16, Rsf1, ASF/SF2, Hrb98DE, Neos, PSI), and all but Prp8 bind RNA preferentially in early embryos, when embryonic development relies on maternally deposited mRNAs. We tested whether the reduced RNA binding of ASF/SF2 and PSI in the 4.5–5.5 h sample could be due to different abundance of their target RNAs at those time points, and calculated expression change of mRNAs identified as their presumable targets in an RNAi screen[33]. The abundance change of splicing factor target mRNAs between 4.5–5.5 h and 0–1 h embryos is statistically indistinguishable for all four splicing factors, and for all genes (Fig. 5b).

To assess whether alternative mRNA splicing, which has the potential to alter protein structure[34], could account for the modulation of RNA-binding activity of some proteins we observed in our experiments, we performed whole-transcriptome sequencing of 0–1 h and 4.5–5.5 h embryos, and analyzed

differential exon usage[35]. Comparison of transcript expression levels measured in two biological replicates (Supplementary Fig. 4a) shows high correlation between replicates. According to our results, mRNAs coding for RNA interactome proteins were strongly enriched in alternatively spliced species compared to all identified mRNAs (Fischer's exact test for count data: $P$ value = 4.204e − 13). Enrichment rates of alternatively spliced mRNAs among those coding for RBPs and RBP candidates were similar in the three aforementioned dynamic classes (Fig. 5c).

We performed enrichment analysis of protein features encoded by alternatively expressed exons in the entire transcriptomes of early and late stage embryos. Several domains were enriched in sequences encoded by all alternative exons, including two known RBDs—KH-1 and RRM-1 (Fig. 5d). In addition, some RBPs are encoded by alternatively spliced mRNAs, and alternative exons contain RBDs. However, we could not find direct evidence for alternative inclusion of RBDs by mass spectrometry, likely due to insufficient depth of the proteomic data. In accordance with previous findings[36], disordered regions were enriched in exons

that were both upregulated and downregulated at the early stage. Interestingly, disordered regions were enriched stronger in the exons that were preferentially used at the early stage ($P$ value = 5.56e − 48) than in the exons used at the late stage ($P$ value = 3.23e − 02) (Supplementary Fig. 4b,c). When compared to exons preferentially used at the late stage, alternative exons preferentially used in early stage mRNAs are enriched for 27 different domains, including some well-characterized RBDs (Supplementary Fig. 4c).

## Discussion

This resource presents the first, to our knowledge, comprehensive list of RBPs in *D. melanogaster* at different developmental stages obtained in an unbiased proteome-wide experiment. Selection of the RNA-bound protein fraction was achieved through cross-linking of RBPs to RNA by ultraviolet light and specific precipitation of poly(A)+ RNA on oligo(dT) magnetic beads under stringent conditions. We improved the RNA capture procedure, that was initially optimized for cell lines[12], by increasing lysis temperature to 60 °C and using higher DTT concentration (12.5 mM). This step allowed the complete removal of tubulin, which is presumably not an RBP, and did not lead to any detectable RNA damage.

Analysis of captured samples by the recently developed TMT MS3 method[17] identified 523 high-confidence RBPs in *Drosophila*, despite the fact that enrichment of some proteins could not be quantified due to missing intensity values in the control noCL sample. Although this technical issue made data analysis challenging, it signified that our efforts in reducing protein background resulted in generation of extremely high quality RBP specimens, where background signal in the negative control was close to zero for most of the proteins quantified in the ultraviolet-irradiated samples. Our data show that TMT MS3 allows efficient reduction of experimental noise, but challenge the current data analysis pipelines where the existence of an intensity ratio between the two conditions is required to quantify a protein[37].

We generated the most comprehensive list of high-confidence *Drosophila* RBPs so far by applying the optimized RNA interactome capture method to embryos at two distinct

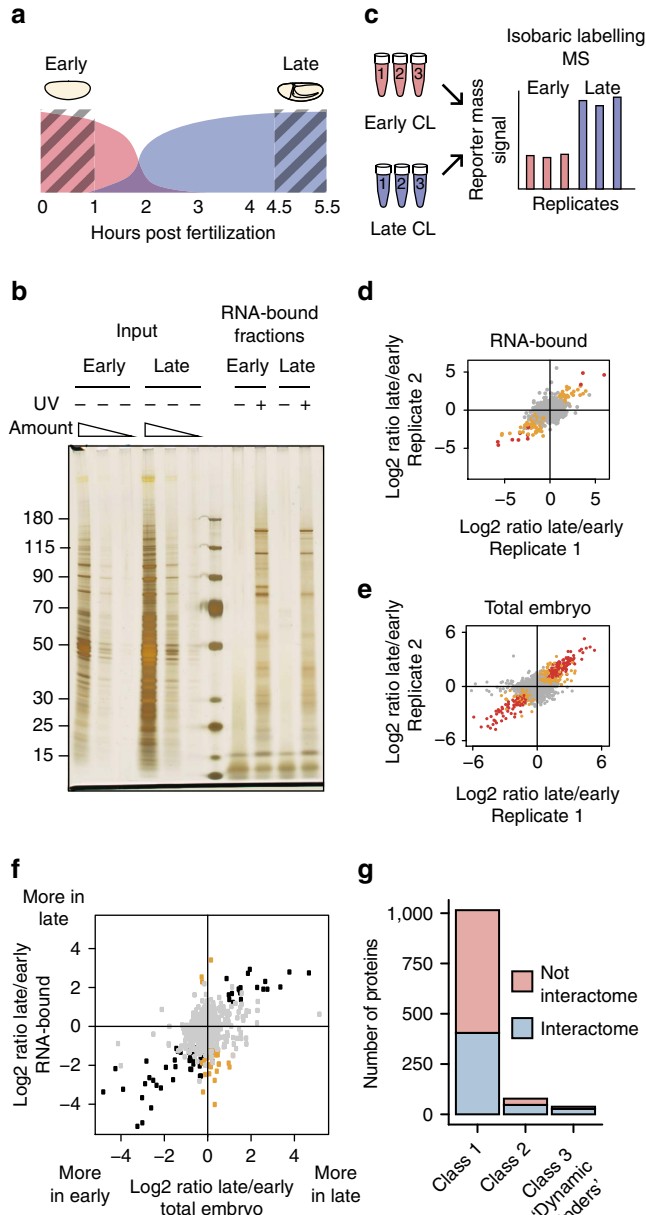

**Figure 4 | Study of RNA interactome dynamics during the maternal-to-zygotic transition.** (**a**) Schematic representation of the MZT. Shaded areas show the time frames considered in our study. (**b**) Proteins captured from three biological replicates of CL early (0–1 h) and late (4.5–5.5 h) embryos were partially digested, labelled by six different TMT labels and analyzed by MS. (**c**) Protein profiles of total lysates and RNA-bound fractions recovered from early and late embryos, noCL and CL. (**d,e**) Ratios of protein abundance late/early in two of the three biological replicates. (**d**) RNA-bound fractions, (**e**) total lysates. Red dots: proteins whose abundance significantly changes at FDR 1%; yellow dots: proteins whose abundance changes at FDR 10%. Supplementary Fig. 3a,c includes the same plots, with $P$ values and Pearson correlation values indicated. (**f**) Scatter plot showing change of average protein abundance early/late in RNA-bound fractions (ordinate) and total protein lysates (abscissa). Grey dots: proteins whose abundance does not significantly change neither in the RNA-bound fraction, nor in the total embryo lysate. Black dots: proteins whose abundance change in the RNA-bound fractions follows the change in the total embryo lysate. Yellow dots: proteins whose abundance in the RNA-bound fraction significantly changes, and is inconsistent with their abundance change in the total protein lysate, suggesting active regulation of their RNA-binding capacity ('dynamic binders'). (**g**) Numbers of proteins in each of the dynamic classes defined in (**f**). Red: numbers of proteins belonging to the RNA interactome. Blue: numbers of proteins not belonging to the interactome.

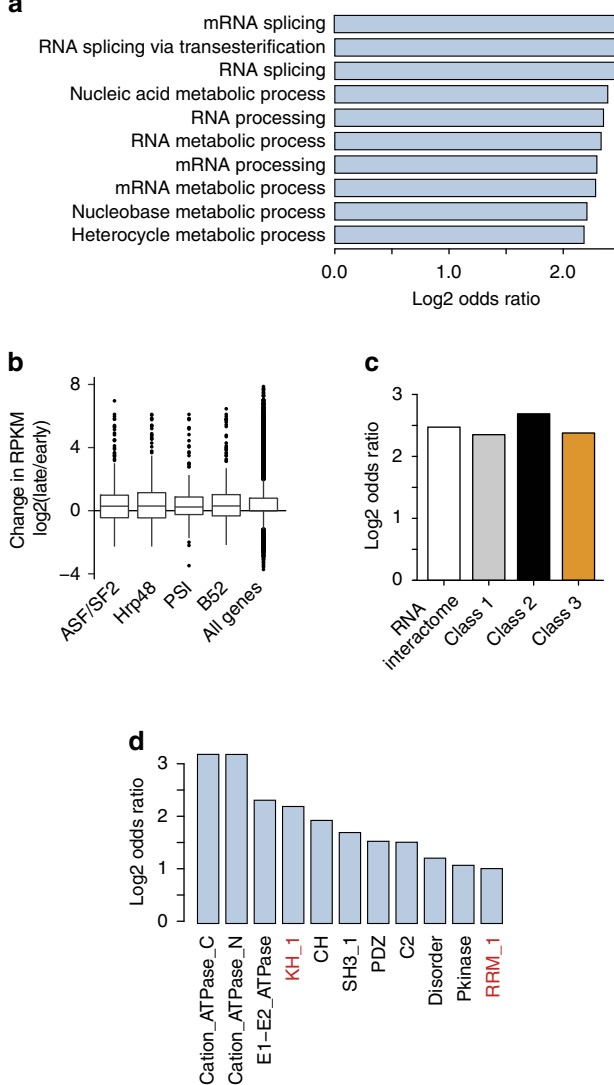

**Figure 5 | Possible explanations for dynamic behaviour of some of the RBPs during the MZT.** (**a**) Ten most enriched GO terms related to biological process among the class 3 RBPs and RBP candidates ('dynamic binders'). (**b**) Log2 change in expression levels (RPKM) of putative transcript targets of four splicing factors and all genes. Splicing factors' targets are identified as transcripts whose abundance is significantly affected ($|z$-score$|>2$) upon depletion of a splicing factor[33]. Comparison of changes in expression of all genes and splicing factor targets using the Student $t$-test resulted in the following $P$ values: ASF/SF2—0.6698; Hrp48—0.1532; PSI—0.5239; B52—0.1847. (**c**) Enrichment of alternatively spliced mRNAs in the whole RNA interactome and in the three dynamic classes. Odds ratios are presented. (**d**) Pfam domains enriched among exons that are differentially used between 0–1 h and 4.5–5.5 h embryos Known RBDs are indicated in red.

developmental stages—before and after the MZT. Our analysis confirms RNA-binding ability of hundreds of previously known RBPs and provides new links between RNA metabolism and development, including new evidence for RBP involvement in regulation of cell differentiation[38]. Some well-known RBPs expressed in the early embryo were not captured in our experiments, possibly because of association with deadenylated mRNAs present at early stages[39], or the limited ability of the 254 nm ultraviolet light to penetrate the optically dense embryo, and therefore, activate nucleobases of RNAs located in its deepest

layers[14]. Our results provide first insight into the molecular function of many essential proteins, including those involved in neurodevelopment, and call for further mechanistic studies.

Our comparison of the *Drosophila* interactome with data available for four other species reveals that proteins that are conserved and appear in all interactomes are predominantly previously known RBPs involved in core processes of RNA metabolism, such as translation and processing, whereas species-specific proteins are enriched for novel RBPs. This also highlights the utility of an experimental technique such as interactome capture for discovery of novel, species-specific RBPs.

In light of evidence indicating that regulation of metabolism might be tightly bound to the MZT in *Drosophila*[26], and previous reports of RNA binding by metabolic enzymes in other organisms[8,10,11], the discovery of such enzymes in *Drosophila* is particularly interesting. It was previously shown that 37 mRNAs encoding metabolic enzymes are bound and possibly downregulated by a key MZT driver, Smaug[26]. RNA binding by metabolic enzymes indicates that an additional post-translational mechanism might regulate their activity. The number of metabolic enzymes capable of binding RNA is lower in *Drosophila* than in other organisms for which RNA interactomes are available. Why this might be, as well as the roles of metabolic enzymes in RNA metabolism and regulation, should be clarified by future studies.

We confirmed the ability of two newly discovered RBPs, CycB and EB1, to bind RNA directly by a reciprocal approach that involves precipitation of an individual protein and detection of the bound RNA by radioactive labelling. In addition, based on the labelling patterns of presumable degradation products of EB1-GFP, we propose that its APC binding domain harbours RNA-binding activity, not excluding, however, that the N-terminal microtubule-binding domain of EB1 may also contribute to RNA binding. In contrast, only the full-length version of GFP-CycB was radioactively labelled, suggesting that either the full-length or the C-terminal fragment of CycB mediates RNA binding.

We propose that RNA binding might have a role in CycB regulation or targeting: for example, localized RNAs might tether CycB to the loci where its activity is required, such as centrosomes[40]. The RNA-binding ability of EB1 might indicate a role in microtubule dependent transport of mRNAs, as was recently proposed for APC, another microtubule end-binding protein also shown to bind RNA[41]. Further experiments will shed light on the biological importance of the RNA-binding activity of CycB and EB1.

Analysis of the RNA interactomes of pre- and post-MZT *Drosophila* embryos provided in this study reveals, expectedly, high dynamicity of the RNA-bound fraction during the MZT and confirms deeper involvement of RBPs in the early, pre-MZT embryo that is transcriptionally inactive. Comparison of protein abundance changes in the RNA-bound fraction and in the total proteome allowed their separation into three dynamic classes, which indicates that various mechanisms might regulate RNA binding, including active regulation of an RBP's affinity for RNA and isoform exchange. Possible active regulation mechanisms include, but are not limited to allosteric regulation by trans-acting factors such as proteins, RNAs or small molecules, competitive binding of trans-acting factors at the RNA-binding site and post-translational modifications. Some RBPs might also appear to bind RNA more efficiently at one of the stages if the mRNAs that they bind have undergone a change in polyadenylation status. Observed dynamic behaviours of some of the known RBPs such as splicing factors indicate that they may have other functions in addition to previously characterized ones. The fact that these splicing factors preferentially bind RNA in the early, pre-MZT

embryo is surprising for two reasons: first, transcription, and accordingly, pre-mRNA processing is generally low in the rapidly dividing nuclei of embryos at the early, syncytial stage[42]. Second, the nucleus–cytoplasm ratio in early embryos is low, hence increased abundance of the splicing factors in the early stage indicates that they are bound to cytoplasmic RNAs. This could reflect that the splicing factors are either stockpiled in an inactive, RNA-bound state until the onset of transcription, or they perform so far unexplored cytoplasmic roles in the early embryo. These findings open new avenues for research and will help better understand the universal process of the MZT.

By analyzing protein features encoded by exons that are differentially expressed during the MZT we determined that alternative splicing indeed has the potential to affect the architecture of an RBP and therefore can impact on the biochemical properties of the protein. The fact that inclusion of some RBDs occurs preferentially in the early, transcriptionally quiescent stage aligns well with the special importance of RBPs in the post-transcriptional control of maternal mRNAs.

Differential exon expression analysis also revealed that disordered regions are alternatively spliced at both stages, in accordance with previous reports[36]. The stronger enrichment of disordered regions in alternative exons upregulated in the early embryo correlates with the broader involvement of RBPs at this stage. Such regions may have various roles in the early embryo: in addition to their presumable binding and regulation of RNA[8,43], they may be involved in assembly of higher order protein or protein–RNA complexes, in processes similar to phase transition and hydrogel formation[44,45], or the binding of proteins to the C-terminal domain of RNA polymerase II (ref. 9) (reviewed by Järvelin et al.[46]). Large molecular complexes held together by interactions of disordered protein domains might be components of even larger units, and have roles in organization of the early embryo syncytial cytoplasm that is not yet subdivided by membranes to form cells. Further in depth mechanistic studies of disordered regions will advance our understanding of their roles in RNA regulation and other cellular processes.

## Methods

**Fly stocks.** Flies were maintained according to standard procedures at 25 °C. *D. melanogaster* Oregon-R flies were used to obtain embryos for RNA interactome capture (Bloomington Stock 5). RNA-binding activity of CycB and EB1 was validated using the endogenously tagged $w^{1118}$; *cycB* FlyTrap line (Bloomington Stock 51568, (ref. 47)), and flies expressing transgenic *Eb1-gfp* under control of the UAS promoter[48] driven by *nos-gal4::vp16* (ref. 49), respectively.

**Embryo collection, cross-linking, oligo(dT) pull-down.** Zero-to-one hour-old embryos collected from ten population cages each containing ~ 10,000 flies were either processed immediately (0–1 h, early embryos) or incubated for 4.5 h and then processed (4.5–5.5 h, late embryos). After collection on a 0.18 mm mesh sieve, embryos were rinsed with distilled water, weighed and the sample was split into two. One half (noCL, control sample) was homogenized in lysis buffer (20 mM Tris–HCl pH 7.5, 500 mM LiCl, 1 mM ethylenediaminetetraacetate—EDTA, $5 g l^{-1}$ Li-dodecylsulphate, 12.5 mM dithiothreitol—DTT). The other half was irradiated with $1.0 J cm^{-2}$ 254 nm ultraviolet light (Stratalinker 1800) on ice under a layer of ice-cold phosphate buffered saline supplemented with 0.1% Tween-20, collected on a sieve, then homogenized in lysis buffer. After lysis embryo extracts were cleared by centrifugation at 15,000*g* for 30 min, diluted to total protein concentration $0.1 mg ml^{-1}$, incubated at 60 °C for 15′ and cooled to room temperature.

Poly(A)+ RNA was captured according to the protocol of ref. 12. A total of 2.5 ml suspension of magnetic oligo(dT) beads were used per 100 ml of embryo extract. Oligo(dT) beads were incubated with the extracts for 1 h at room temperature, then washed 1 × with lysis buffer, 2 × with buffer containing 20 mM Tris–HCl pH 7.5, 500 mM LiCl, 1 mM EDTA, $1 g l^{-1}$ Li-dodecylsulphate, 1 mM DTT, 2 × with buffer containing 20 mM Tris–HCl pH 7.5, 500 mM LiCl, 1 mM EDTA, 1 mM DTT, 2 × with buffer containing 20 mM Tris–HCl pH 7.5, 200 mM LiCl, 1 mM DTT. RNA-protein complexes were eluted by vigorous shaking at 55 °C in elution buffer (20 mM Tris–HCl, 1 mM EDTA).

To obtain optimal yields of poly(A)+ RNA, each lysate was put through the RNA interactome capture procedure four times. Eluates from subsequent rounds of RNA interactome capture were pooled and concentrated using Amicon filter units with a molecular weight cut-off of 3,000 Da.

**Staging of embryos.** An aliquot of embryos used for RNA interactome capture was dechorionated by bleach treatment (30 s incubation in 5% NaClO), rinsed with distilled water and fixed in a 1:1 mixture of heptane and methanol. After gradual rehydration, embryos were stained with $0.1 \mu g ml^{-1}$ 4′,6-diamidino-2-phenylindole (DAPI) and visualized on a confocal fluorescent microscope.

**Quality control of captured RNA complexes.** To control the quality of the captured RNA, an aliquot of each eluate was treated with protease K and subsequently purified as in ref. 12. A total of 100 ng of each RNA sample was analyzed on an Agilent Bioanalyzer 2100 according to the manufacturer's recommendations.

For quantification of 18S rRNA and *gapdh1* and *ts* mRNAs, we performed reverse transcription followed by RT-qPCR. RT was performed using the SuperScript III first-strand synthesis system (Thermo Fischer Scientific), equal amounts of total and oligo(dT) captured RNA were used. Dilutions of first-strand DNA libraries prepared from total noCL RNA were used as standards. Amplification was performed in the presence of 1 mM of each oligonucleotide and 1 × SYBR Green RCR Master Mix (Applied Biosystems). Relative amounts of targets were calculated using StepOne software. Sequences of oligonucleotides used for quantification of 18S rRNA were taken from ref. 50. Sequences of oligonucleotides used for quantification of *gapdh1* and *ts* mRNAs were designed with Primer3 software[51,52]. Sequences of oligonucleotides used in the study are presented in the Supplementary Table 1.

**Analysis of captured proteins.** Eluates were supplemented with RNase buffer to a final concentration 20 mM Tris–HCl pH 7.5, 150 mM NaCl and $200 U ml^{-1}$ of RNases A and T1. After 30 min incubation at 37 °C an aliquot of proteins was supplemented with LiDS loading buffer (Thermo Fischer Scientific) containing 20 mM DTT, separated on 4–12% polyacrylamide gels (NuPage, Thermo Fischer Scientific) and either stained with silver using Pierce Silver Stain Kit (Thermo Fischer Scientific) or electrophoretically transferred onto polyvinylidene fluoride (PVDF) membranes.

PVDF membranes were blocked in phosphate buffered saline supplemented with 0.1% Tween-20 (PBST) and containing $100 g L^{-1}$ milk powder. Membranes were then hybridized with antibodies diluted in PBST-milk that are listed in Supplementary Table 2. Membranes were washed with PBST, hybridized with secondary antibodies conjugated with horseradish peroxidase (GE Healthcare) and developed using Western Lightning Plus ECL kit (PerkinElmer). Full-size images of the most important immunoblots are available in the Supplementary Fig. 1.

**MS, protein identification and quantification.** Captured proteins were digested with Lys-C and labelled with 6plex TMT as previously described[53], fractionated by high pH reverse phase chromatography and analyzed on LTQ-Orbitrap Velos Pro mass spectrometer (Thermo Scientific). MS analysis and identification and quantification of proteins is described in detail in Supplementary Note 2.

**Validation of RNA-binding ability by PNK assay.** Zero-to-two hour-old embryos expressing GFP, GFP-CycB or EB1-GFP and $w^{1118}$ embryos were collected and irradiated with $1 J cm^{-2}$ ultraviolet light (254 nm) and lysed in 10 × RIPA buffer (20 mM Tris–HCl pH 7.5, 150 mM KCl, 0.5 mM EDTA, 1% Triton-X100, 1% deoxycholate and 0.1% sodium dodecylsulphate—SDS, 5 mM DTT, 5 × Roche protease inhibitor cocktail) and cleared by 10 min centrifugation at 15,000*g*. Lysates containing GFP were diluted with $w^{1118}$ lysates to reduce GFP concentration to levels comparable with those of GFP-CycB and EB1-GFP. Lysates were diluted with RIPA buffer to $10 mg ml^{-1}$ total protein, then diluted 1:10 with low salt buffer (20 mM Tris–HCl pH 7.5, 150 mM KCl, 0.5 mM EDTA) to reduce the concentration of detergents. A total of 10 µl of suspension of magnetic beads coated with *Lama alpaca* anti-GFP antibody (Chromotek GFP-Trap_M) were added to 1 ml of each lysate and incubated at 4 °C for 2 h. Beads were washed six times with 1 ml high salt buffer (20 mM Tris–HCl pH 7.5, 500 mM KCl, 1 mM EDTA, 0.1% SDS, 0.5 mM DTT, 1 × Roche protease inhibitor cocktail). Washes were performed at room temperature to avoid SDS precipitation. Beads were then washed twice with 1 × T4 PNK buffer B (Thermo Fischer Scientific) and incubated for 15 min in 10 ul of reaction mix containing 1 × T4 PNK buffer B, 2 µl T4 PNK and 1 mCi γ-$^{32}$P-adenosyntriphosphate (γ-ATP) (Hartmann Analytic). Beads were then separated from the reaction mix and washed 5 times with low salt buffer supplemented with 0.1% Triton-X100. Labelled complexes of GFP proteins and RNA were eluted by incubation in 1 × LiDS buffer (Thermo Fischer Scientific), separated on polyacrylamide gels and transferred to PVDF membranes as described above. Autoradiography was recorded using phosphor screens and GE Typhoon 7000 scanner, and GFP proteins were detected as described above.

**Transcriptome analysis.** To determine the total transcriptome, poly(A)+ RNA from 0–1 h and 4.5–5.5 h-old embryos was purified as described above, treated

by Turbo DNase I (Thermo Fischer Scientific) according to manufacturer's instructions and purified by phenol–chloroform extraction and ethanol precipitation. cDNA libraries were prepared form the resulting samples using a SENSE mRNA-Seq Library Prep Kit V2 (Lexogen) and analyzed by single-end 50 sequencing on an Illumina HiSeq2000. Reads were aligned to the *Drosophila* reference genome version R6.07 using STAR. Differential exon usage was called using the R/Bioconductor package DEXSeq.

**Data availability.** The MS proteomics data that support the findings of this study have been deposited in the ProteomeXchange Consortium via the PRIDE partner repository[54] with the dataset identifier PXD003882. All relevant data and computer code that support the findings of this study are available from the authors upon request.

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

## Acknowledgements

We thank Anne-Marie Alleaume, Benedikt Beckmann, Rastislav Horos and the rest of the Hentze lab (EMBL) for sharing their expertise on RNA interactome capture and PNK assay. We are grateful to Aleš Obrdlik for advice on RNA interactome capture optimization and the rest of the Ephrussi lab for useful discussions. We are grateful to Matt Rogon and the Center for Statistical Data Analysis at EMBL, and Lin Gen for their help with computational data analysis, to the Genomics Core Facility at EMBL for performing transcriptome sequencing and expert advice, and to Sofia Föhr and the Proteomics Core Facility at EMBL for performing mass spectrometry and for technical

expertise and support. We are grateful to Donald Rio and Akira Nakamura for their gifts of antibodies. V.O.S. is supported by the EMBL International PhD Program and C.K.F. by an EMBO postdoctoral fellowship (LTF 1006-2013). B.F. is supported by the Helmholtz Association (VH-NG-1010). A.C. is supported by an MRC Career Development Award (MR/L019434/1). This work was funded by the European Molecular Biology Laboratory. M.W.H. acknowledges support by the ERC Advanced Grant ERC-2011-ADG_20110310.

## Author contributions

V.O.S., A.C. and A.E. designed the experiments. V.O.S. performed the experiments and led the data analysis. C.K.F. and J.K. analyzed the samples by mass spectrometry. C.K.F., B.F. and I.G. analyzed the data, and all authors discussed the data. V.O.S., A.E. and A.C. drafted the manuscript. V.O.S. wrote, and all authors contributed to the revision of the manuscript.

## Additional information

**Competing financial interests:** The authors declare no competing financial interests.

