## [Peer review file · Nature Communications]

Reviewers' Comments:

Reviewer #1 (Remarks to the Author)

The authors adapted a recently developed method of RNA-interactome capture to systematically analyse the proteins binding to polyadenylated RNA (mainly mRNA) in *Drosophila melanogaster* during maternal-to-zygotic transition (MZT). Although the technique has been previously used in different model organisms, such as unicellular yeast and multicellular nematodes *C. elegans*, the authors further optimized the protocol for this first *Drosophila* study and further extended on studying the dynamics of RNA-protein interaction at two different developmental stages. Using the approach, they identified 528 mRNA-binding proteins, and two new ones were further validated with a biochemical approach. Perhaps more interesting, they applied a multiplex proteomics approach comparing changes of the RNA-protein interaction with that of overall protein levels (proteome) across the 2 developmental stages. Although changes in the mRNA-protein interactome were largely reflected at the protein level (group 1), a small group of proteins exclusively changed mRNA binding (group 3, 38 proteins). It is interesting that this group is enriched in splicing factors. Further integration with transcriptomic data suggests that alternative splicing of these proteins could involve the addition/exclusion of exons coding for RNA-binding domains or unstructured regions, which is very interesting.

Specific comments to be addressed in the order appearing in the manuscript:

The authors have previously reported the presence of metabolic enzymes and other "enigmaRBP" in yeast and human RNA-protein interactomes. I was wondering whether the *Drosophila* interactome also includes metabolic enzymes? The authors should discuss this point.

1. Page 5, Figure S1a. Irradiation of *Drosophila* embryos with UV. I am a bit confused why the authors conclude that 1 J/cm² of UV irradiation is the best choice? It seems that 2 J/cm² leads to higher protein yields (although not very comparable since they are part of different gels). Looking at the inputs, an increase in most of the proteins can be seen after the crosslinking but unfortunately, no data shown for 2 J/cm². Quantification of the Western Blots analysis to support their argument for choosing the dose would be good. Fig S1b, the authors may want to label the peaks referring to ribosomal RNAs at ~2,000 nts.

2. Extract treatment at 60 degrees: I am not fully convinced about the advantages of the 60 degree treatment of the extracts as the data shown in Figure 1e are not conclusive. First, the input levels and eluates of the samples without treatment (-) are generally higher than the treated ones (+) fractions. Secondly, the recovery of control RBPs is also substantially lower in temp treated samples (respective bands for PABP, KHC and eIF4E are much weaker in + samples. Since the tubulin band is already weak in the (-) samples, the absence of the tubulin band in + sample may simply reflect less recovered material but is no indication for improved specificity (e.g. using equally downsized input material may likewise lead to a disappearance of the tubulin band). Furthermore, the heat treatment may lead to degradation of proteins - showing an uncut gel may disprove this (in the Supplement).

3. MS analysis. Authors should provide the full dataset of their analysis in the supplement. Supplementary File 1 indicates the replicate 1 - 3 (e.g. logRatio.Replicate1) but it is not clear what this replicate refers to? Is this data for early or late MZT or both? Thus, the authors should present full raw and processed data for all 12 samples analysed as depicted in Fig. 1f. Moreover, MS data should be deposited at MS database and accession numbers provided. The columns and their meaning in all the Supplementary Files should be better explained in the associated description of Supplementary Files. Furthermore, the missing value analysis should be better described as this semiquantitative analysis contributes to the most of the selected potential mRBPs.

4. How reproducible is the MS data? Authors should add the correlation analysis (r-value and p-values) for all samples currently shown (log CL/noCL)(Fig. 1g). Since the semiquantitative analysis contributes to most proteins of the interactome, it would even be more important to show the correlation among CL (triplicates) and corresponding the non-crosslinked samples to estimate the reproducibility of non-transformed data. The author may think about some graphical representation of their dataset.

5. Page 8 and page 10. I have problems with this kind of data representation, both for the description of novel RBPs in *Drosophila* and also in the comparison of RBPs across different species. It is hard to find what the numbers detailed in the text refer in the figures. In page 8 I do not find in the figure 'the number of such proteins was 293 (out of 2655, 8%)...', neither 'the total embryo proteome was 639 (out of 3655, 17%)...'.

6. Domain analysis - Page 9, Fig. S2. Except for RRM, ribosomal and KH domains, there is lack of enrichment for other well-described canonical RBDs (e.g. dsRBP, Zinc-finger domains, etc.). The authors should discuss this issue and mention it in the results/discussion.

7. Disordered regions, Page 9. The observation could be supported by more rigid statistical analysis. Are the motifs depicted in Fig. 2d significantly overrepresented among proteins of the experimentally defined mRNA-protein interactome? Moreover, the graph in figure 2c is not sufficiently described yet. What do the "novel RBPs" (purple line) refer to? X-axis should be labeled and P-values added or described in text. Interestingly, a similar pattern of motifs has been found in mouse embryonic stem cells (less enriched in disordered regions), the readers wonders whether this related to a special feature of RBPs in embryo stage, irrespective of the organism? Moreover, since likewise motifs have been seen in yeast and *C. elegans*, the reader may be interested to see how it compares with these organisms.

8. Interactome comparison, pg 10, Fig. 2e. The authors state that they compared the interactomes of human, mouse, worm and yeast. However, the Venn diagram of Fig 2e only depicts overlap with human and mouse datasets. The authors should take the opportunity and provide a more profound discussion of data comparison in light of evolutionary conservation, which will certainly be of interest to the reader.

9. Localization data, pg. 11. Interesting to see the relation to neuroblast, however the current presentation of the data is not very enlightening. Is there a statistically relevant overrepresentation of localized proteins?

10. Labeling and legend for figure 3 need revision and editing. If applicable, the authors should quantify data to get an idea what fraction of cycB or EB1 is bound to RNA in cells?

11. The presence of many splicing factors among the group 3 proteins is intriguing and very interesting. Moreover, the observation that alternative exon often code for RBDs or disordered region. The authors should try to further consolidate this finding - for instance, they could look at their MS data for the 38 proteins - is there direct evidence for the presence of the alternative spliced forms?

12. Discussion pg 17. The argument that TMT MS2 allows efficient reduction of experimental noise seems a bit far reached. At the end, the semi-quantitative approach was beneficial to come-up with a list of mRBPs. TMT is suitable for comparing treated and untreated samples, however without strict further validation of the data - no judgement on the performance of this versus other approaches can be made in my opinion.

Reviewer #2 (Remarks to the Author)

In this manuscript, Sysoev et al provide the first global in vivo identification of RBPs in *Drosophila*.

They achieve this using the "RNA interactome capture" method first developed by Hentze and colleagues in human tissue culture cells, which has also been used to identify RBPs in mouse, *C. elegans*, and yeast. The identification of *Drosophila* RBPs presented here complements these previous studies and provides additional insights into the conservation of RBPs in eukaryotes. Moreover, the work presented here provides an assessment of the dynamics of the RNA-binding proteome during a developmental transition - the maternal-to-zygotic transition during *Drosophila* embryogenesis (although see caveats listed below).

The experiments and analysis appear generally sound (again, see caveats listed below), and the list of *Drosophila* RBPs obtained represents an important resource. However, various technical aspects of the study require further clarification, additional analyses should be performed to more thoroughly assess the properties of the identified RBPs, and several caveats need to be included when presenting the data and conclusions, as described below:

1) 254nm UV irradiation penetrates only a short distance into *Drosophila* embryonic cytoplasm (see for example, Togashi and Okada 1983 *Dev Growth Diff*). Are the authors sure that their UV treatment crosslinked RBPs to RNAs throughout the embryo, as opposed to mostly near the periphery? If not this caveat must be mentioned.

2) The need to rely on the polyA tail for RBP capture limits the ability to capture all RBPs at any individual stage, and in particular is problematic for the purposes of comparing the RNA-bound proteome at early and late timepoints during the maternal-to-zygotic transition. Changes in tail length are a major feature of this transition with regard to the regulation of mRNA decay and translational control. Oligo-dT selection will preferentially purify mRNAs with long (rather than short) polyA tails. Thus, the population of mRNAs purified and therefore the associated RBPs is biased. Furthermore, if RBPs are involved in regulating tail length, and bind to targets with different length tails at the early versus late timepoints, this may appear artificially as a change in the RBP's binding activity. This possibility is particularly relevant with regard to the class 3 "dynamic binders" identified by the authors. The authors need to include discussion of these caveats both in the Results and Discussion.

3) I am confused as to how the authors used the different timepoints in their initial definition of the RNA interactome. My understanding is that they were treated as pooled samples at this stage to define the interactome, and separate samples later on for assessing the dynamics of RNA binding. The authors should explain these details more clearly.

4) GO term enrichments among the entire interactome, as well as just the set of novel RBPs, would be informative and should be assessed. For instance, do the authors see an enrichment for metabolic enzymes as has been observed in the RNA-binding proteome of other species? Are there additional enriched functions?

5) What GO terms were considered as "RNA-related"? This should be indicated somewhere.

6) How do novel versus previously characterized RBPs breakdown in terms of differential expression/RNA-binding at the early and late timepoints?

7) Do any of the class 3 RBPs (putative "dynamic binders" - see above) show differential inclusion of RBDs at the early and late timepoints as a result of alternative splicing, based on the sequencing analysis? If not, there is only weak evidence for the authors' assertion that differential splicing is a major contributor to the observed "dynamicity" of RNA binding. Other potential mechanisms should be discussed in more detail, (including the potential biases introduced by the use of polyA-based capture, as discussed above).

Minor comments:

Figure 1c: x-axis labels should be explained more clearly in legend.

What does the asterisk in Figure 1d indicate?

Are the first six lanes in Figure 1d and 1e serial dilutions? This should be explained somewhere.

Supplemental Figure S1, panel (a): again, are the input lanes serial dilutions at the two different UV treatments? This should be indicated.

Figure 1h: what does "B-H" stand for?

Figure 2g: are these differences statistically significant?

Figure 4g: typo in graph label - "interactomle"

Figure 5b: why is there no median indicated in the box plot for "all genes"? Also, the P-value cut-off and statistical test used for the comparison should be indicated.

Reviewer #3 (Remarks to the Author)

The manuscript by Sysoev and colleagues provides a nice repository of RNA binding proteins of early *Drosophila* embryos, and their differential binding to RNA at two developmental stages where post-transcriptional regulation is pervasive. Perhaps not surprisingly, the authors find dynamic RNA binding between the two stages, explained mainly (but not only) by differential protein abundance. Interestingly, a group of proteins enriched in splicing factors shows dynamic binding without changes in protein or target abundance, suggesting regulated binding to RNA. Finally, the method described for comparative interactome capture will be useful for others attempting similar experiments in other settings.

I have only some minor comments:

1) Some gels/graphics are not well labeled. For example, in Figs 1d-e and 3a do the several lanes for "input" or "unbound" represent decreasing amounts of material? In Fig 3a, please indicate the positions of the fusion proteins alluded to in the text. In Fig 2c, please label the X axis.

2) In Fig 3d the n° lanes in the autoradiography and the Western blot is not the same. Are these the same gels?

3) The authors use 1 J/cm² as the energy of crosslinking, but Fig S1a shows a better crosslinking efficiency at 2 J/cm². Did the authors find RNA degradation at this energy?

4) Page 10, last lane: I guess the authors mean Figure 1f (not 1b)

5) Page 8: The numbers for the total embryo proteome do not coincide with those shown in Fig 2a.

6) Page 42, first paragraph: Please, explain better the FDR threshold that was selected to consider proteins not changing in abundance.

Response to Referees

We thank the reviewers for their careful assessment of our manuscript. We are pleased that they found our resource valuable and are grateful for the constructive comments and suggestions, which we have taken to heart in our revision. Below please find our point-by-point responses to the comments and a description of the changes we have implemented in the revised manuscript and figures. All changes in the manuscript are in red, with the exception of minor edits introduced to comply with length requirements (for instance in the headings).

Reviewer #1:

The authors adapted a recently developed method of RNA-interactome capture to systematically analyse the proteins binding to polyadenylated RNA (mainly mRNA) in *Drosophila melanogaster* during maternal-to-zygotic transition (MZT). Although the technique has been previously used in different model organisms, such as unicellular yeast and multicellular nematodes *C. elegans*, the authors further optimized the protocol for this first *Drosophila* study and further extended on studying the dynamics of RNA-protein interaction at two different developmental stages. Using the approach, they identified 528 (there might be a typo – our interactome contains 523 high confidence hits – *authors*) mRNA-binding proteins, and two new ones were further validated with a biochemical approach. Perhaps more interesting, they applied a multiplex proteomics approach comparing changes of the RNA-protein interaction with that of overall protein levels (proteome) across the 2

developmental stages. Although changes in the mRNA-protein interactome were largely reflected at the protein level (group 1), a small group of proteins exclusively changed mRNA binding (group 3, 38 proteins). It is interesting that this group is enriched in splicing factors. Further integration with transcriptomic data suggests that alternative splicing of these proteins could involve the addition/exclusion of exons coding for RNA-binding domains or unstructured regions, which is very interesting.

Specific comments to be addressed in the order appearing in the manuscript:

The authors have previously reported the presence of metabolic enzymes and other "enigmaRBP" in yeast and human RNA-protein interactomes. I was wondering whether the *Drosophila* interactome also includes metabolic enzymes? The authors should discuss this point.

Indeed, our *Drosophila* RNA interactome includes many enzymes, including several metabolic enzymes. A total of 47 of the 268 proteins annotated as enzymes (Expasy enzyme database, 13-April-2016 release; <http://enzyme.expasy.org/>) were recovered as high-confidence RBPs in our experiments and are thus included in our RNA interactome. In the revised manuscript, we have added a Supplementary File 3 that lists all *Drosophila* enzymes (Expasy), indicating (a) whether they are part of the RNA interactome, (b) whether they have orthologs in mammals, yeast, or worm and, if so, (c) whether the orthologs are part of the RNA interactome of the respective organism. We also include a brief description and a discussion of

these findings in the Results and Discussion sections of the revised manuscript.

Results (page 12): “Discovery of several metabolic enzymes as RBPs in *Drosophila* is consistent with findings of other interactome capture studies¹⁻³ and is noteworthy as energy production was shown to be an important factor in developmental processes in some organisms^{4,5}, and its regulation might play a role in the MZT in *Drosophila*⁶. As for the mammalian and yeast interactomes², our *Drosophila* interactome contains a substantial number (47) of proteins listed as enzymes in the ExPASy database (<http://enzyme.expasy.org/>). Whereas in yeast all steps of the glycolytic pathway can be catalyzed by enzymes identified as RNA binders², in *Drosophila* only two glycolytic enzymes – phosphofructokinase and phosphoglyceromutase – are found in the RNA interactome. Further experiments on individual proteins will elucidate the role(s) of RNA binding by metabolic enzymes in development.”

Discussion (page 20): “In the light of evidence indicating that regulation of metabolism might be tightly bound to the MZT in *Drosophila*,⁶ and previous reports of RNA binding by metabolic enzymes in other organisms¹⁻³ the discovery of such enzymes in *Drosophila* is particularly interesting. It was previously shown that 37 mRNAs encoding metabolic enzymes are bound and possibly downregulated by a key MZT driver, Smaug⁶. RNA binding by metabolic enzymes indicates that an additional post-translational mechanism might regulate their activity. The number of metabolic enzymes capable of binding RNA is lower in *Drosophila* than in other organisms for which RNA

interactomes are available. Why this might be, as well as the roles of metabolic enzymes in RNA metabolism and regulation, should be clarified by future studies.”

1. Page 5, Figure S1a. Irradiation of *Drosophila* embryos with UV. I am a bit confused why the authors conclude that 1 J/cm² of UV irradiation is the best choice? It seems that 2 J/cm² leads to higher protein yields (although not very comparable since they are part of different gels). Looking at the inputs, an increase in most of the proteins can be seen after the crosslinking but unfortunately, no data shown for 2 J/cm². Quantification of the Western Blots analysis to support their argument for choosing the dose would be good. Fig S1b, the authors may want to label the peaks referring to ribosomal RNAs at ~2,000 nts.

First, we should clarify that the panels shown in the original Supplementary Fig. 1a show samples that were run on the same gel and are parts of the same blot and image, and therefore can be compared. We removed a lane (between the 2 panels), because material in that sample was lost either during the experiment or loading. To avoid confusion, we have replaced the original (split) image with a different one (see revised Supplementary Fig. 1a).

We selected 1.0 J/cm² as the minimum UV dose required to promote efficient protein RNA crosslinking without causing RNA degradation in embryos. Although irradiation with 2.0 J/cm² in some instances resulted in slightly higher crosslinking efficiency, occasionally it also resulted in tubulin

contamination of the UV cross-linked samples (see new Supplementary Fig. 1a). Additionally, in order to achieve a dose of 2.0 J/cm², the duration of UV exposure had to be doubled (compared with 1.0 J/cm²). Assuming that shorter embryo processing times would favor a higher quality of the lysates, and given that UV irradiation with 1.0 J/cm² was sufficient to precipitate enough protein for MS analysis, we opted for the lower dose. Finally, the shorter time enabled us to generate multiple samples (all steps, from embryo collection to obtention of lysates) in one embryo collection day. This is crucial, given that the *Drosophila* embryo collection cages are populated with adult flies and produce embryos only a few days per month. For these reasons we considered the UV dose of 1.0 J/cm² to be a good compromise between irradiation time and crosslinking efficiency. In the revision, we have added this information as Supplementary Note 1.

As recommended by the reviewer, we have labeled rRNA peaks (Fig. 1b and Supplementary Fig. 1g).

2. Extract treatment at 60 degrees: I am not fully convinced about the advantages of the 60 degree treatment of the extracts as the data shown in Figure 1e are not conclusive. First, the input levels and eluates of the samples without treatment (-) are generally higher than the treated ones (+) fractions. Secondly, the recovery of control RBPs is also substantially lower in temp treated samples (respective bands for PABP, KHC and eIF4E are much weaker in + samples. Since the tubulin band is already weak in the (-) samples, the absence of the tubulin band in + sample may simply reflect less recovered material but is no indication for improved specificity (e.g. using

equally downsized input material may likewise lead to a disappearance of the tubulin band). Furthermore, the heat treatment may lead to degradation of proteins - showing an uncut gel may disprove this (in the Supplement).

In the experiment shown in the original Fig. 1e, input amounts of extracts either treated or not treated at 60°C were indeed slightly different. However, we do not think the input differences explain the difference in amounts of oligo(dT)-captured proteins from extracts exposed or not to 60°C, because (a) equal amounts of captured RNA were loaded on the gel, and (b) the difference in the amounts of captured eIF4E in treated vs. untreated samples is much greater than that of the respective inputs. We hypothesize that a significant fraction of control RBPs captured from the non-treated lysates was in fact not covalently bound to RNA, yet co-purified due to partial solubilization of the proteins during the lysis procedure. This is supported by the fact that heat treatment in the presence of a 12.5mM DTT results in complete removal of non-covalently bound proteins, possibly due to disruption of any remaining S-S bridges, while genuine RBPs cross-linked to RNA remained on the beads.

For greater clarity, in the revised manuscript we have replaced the data in the original Fig. 1e with the results of another experiment, in which input concentrations were equal (see revised Fig. 1e). However, in this experiment, the benefits of 60°C treatment are not obvious, as neither of the samples contained detectable amounts of tubulin. Taken together, these two examples – the original Figure 1e (now Supplementary Fig. 1h) and the revised Fig. 1e – highlight the variability of contamination that we observed in the absence

incubation of the lysates at 60°C and 12.5 mM DTT. For this reason, all subsequent experiments in our interactome capture study were performed under the more stringent conditions. Full-size blots of the revised Fig. 1e are presented in the revised Supplementary Fig. 1i-p, which shows that heat treatment did not induce protein degradation.

Finally, we would like to mention that our decision to introduce heat treatment of the lysate was based on a number of experiments aimed reducing contamination using other approaches. We first hypothesized that the high protein concentrations in embryo lysates might cause proteins to non-covalently stick to the beads, and therefore performed RNA interactome capture using more diluted lysates and therefore with lower protein concentrations (0.1 mg/ml instead of 1 mg/ml). Although this approach resulted in a moderate improvement, it did not allow complete removal of the indirect RNA binder tubulin from CL samples. We next performed purification of polyadenylated RNA in two sequential rounds of oligo (dT) capture. While the eluates were completely devoid of tubulin and rRNAs, the two components we used to track contamination, total RNA and protein yields, were also dramatically reduced. These attempts, to which we now refer as data not shown (revised manuscript page 6) did not achieve the desired removal of tubulin. Therefore, we settled for heat treatment (in the presence of 12.5 mM DTT), as it presented an ideal balance between stringency and efficiency of capture, as confirmed by mass spectrometry analysis and additional controls of derived samples.

3. MS analysis. Authors should provide the full dataset of their analysis in the Supplement. Supplementary File 1 indicates the replicate 1 - 3 (e.g. logRatio.Replicate1) but it is not clear what this replicate refers to? Is this data for early or late MZT or both? Thus, the authors should present full raw and processed data for all 12 samples analysed as depicted in Fig. 1f. Moreover, MS data should be deposited at MS database and accession numbers provided. The columns and their meaning in all the Supplementary Files should be better explained in the associated description of Supplementary Files. Furthermore, the missing value analysis should be better described as this semiquantitative analysis contributes to the most of the selected potential mRBPs.

We have added the requested data to the supplementary files, including a detailed description of the data listed in the columns. The mass spectrometry proteomics data have been deposited to the ProteomeXchange Consortium via the PRIDE partner repository ⁷, with the dataset identifier PXD003882. This is mentioned in the “Online Methods” section. Reviewers may access the data using the following ProteomeXchange account:

Username: reviewer13520@ebi.ac.uk

Password: m5diEYxC

For greater clarity, we have revised the description of the semi-quantitative analysis in Supplementary Note 2:

“In the case of the RNA interactome, quantitative analysis could only be performed for a low number of proteins because of lack of values in the noCL control, due the low background. Therefore, a second, semi-quantitative approach was applied assuming that peptides without quantitative information are below the detection limit. The number of replicates in which a peptide had been identified was used as a semi-quantitative measure. In total this allows classification of peptides into 16 different groups, as represented in Supplementary Fig. 1

The FDRs were estimated as ratios resulting from division of the transposed matrix in Supplementary Fig. 1 by itself. The following example illustrates this approach. There are 160 peptides that occur in two CL replicates and one noCL replicate, and 7 peptides that occur in one CL and two noCL replicates. FDR for the aforementioned 160 peptides is estimated as $7/160 = 0.04375$. Since only peptides for which $FDR < 0.01$ were considered high confidence hits (green cells Supplementary Fig. 1v), the aforementioned 160 peptides were not considered high confidence hits. Only proteins comprising peptides with $FDR < 0.01$ were included in the *Drosophila* RNA interactome.”

4. How reproducible is the MS data? Authors should add the correlation analysis (r-value and p-values) for all samples currently shown (log CL/noCL)(Fig. 1g). Since the semiquantitative analysis contributes to most proteins of the interactome, it would even be more important to show the correlation among CL (triplicates) and corresponding the non-crosslinked

samples to estimate the reproducibility of non-transformed data. The author may think about some graphical representation of their dataset.

A matrix of scatter plots comparing all the replicates for the quantitative analysis is included in the manuscript (Fig. 1g and Supplementary Fig. 1t). As requested by the reviewer, in the revised manuscript we have added Pearson correlation coefficients and p-values to all such plots (Fig. 1g, Supplementary Fig. 1t and Supplementary Fig. 4). One should not be misled by the lack of significant correlation between CL/noCL ratios of replicate 2 and 3 shown in Supplementary Fig. 1v, as the ratios represent only a small number of detected proteins due to the absence of noCL values, as discussed above (see Comment 3).

The data relevant for the semi-quantitative analysis are shown in Supplementary Fig. 1h. In our opinion, the current representation as a table with colored cells may be more useful than a three-dimensional bar plot, and therefore we did not include an additional graphical representation of these data. As mentioned above (Comment 3), for greater clarity we have revised the description of the semi-quantitative analysis (Supplementary Note 2 and above).

5. Page 8 and page 10. I have problems with this kind of data representation, both for the description of novel RBPs in *Drosophila* and also in the comparison of RBPs across different species. It is hard to find what the numbers detailed in the text refer in the figures. In page 8 I do not find in the

figure 'the number of such proteins was 293 (out of 2655, 8%)...'; neither 'the total embryo proteome was 639 (out of 3655, 17%)...'.

The numbers presented as “total embryo” in the original Fig. 2a represent the total proteome minus the interactome proteins, and therefore do not add up to the total of 3655 proteins identified in the total proteome. We have revised the labeling in the revised figure Fig. 2a, such that this is now clearly stated. It is important to note that not all interactome proteins were identified in the total proteome. We have also added a scale bar indicating the fraction of proteins (left of panel). We apologize for the confusion.

6. Domain analysis - Page 9, Fig. S2. Except for RRM, ribosomal and KH domains, there is lack of enrichment for other well-described canonical RBDs (e.g. dsRBP, Zinc-finger domains, etc.). The authors should discuss this issue and mention it in the results/discussion.

We noted that some RBPs are overrepresented but not significantly enriched according to our statistical threshold. We have included additional examples to the corresponding text on page 9, which now appears as follows:

“Note that most previously known RBDs, such as the La domain, dsRBP (DSRM) and zinc-finger domains, were overrepresented in the RNA interactome compared to the whole embryo lysate but their enrichment did not qualify as statistically significant due to the small number of proteins containing such domains in the *Drosophila* embryo proteome (Supplementary Fig. 2b).”

7. Disordered regions, Page 9. The observation could be supported by more rigid statistical analysis. Are the motifs depicted in Fig. 2d significantly overrepresented among proteins of the experimentally defined mRNA-protein interactome? Moreover, the graph in figure 2c is not sufficiently described yet. What do the "novel RBPs" (purple line) refer to? X-axis should be labeled and P-values added or described in text. Interestingly, a similar pattern of motifs has been found in mouse embryonic stem cells (less enriched in disordered regions), the readers wonders whether this related to a special feature of RBPs in embryo stage, irrespective of the organism? Moreover, since likewise motifs have been seen in yeast and *C. elegans*, the reader may be interested to see how it compares with these organisms.

Throughout the text, when we refer to novel (newly discovered) RBPs we refer to proteins that a) are high confidence interactome hits, b) are not annotated with the GO term "RNA binding" and c) lack a previously described RNA binding domain (e.g. RRM). We define the term "novel RBPs" in the section "Discovery of hundreds of novel *Drosophila* RNA-binding proteins", page 8. As requested, we have labeled the x-axes to Fig. 2c and listed the p-values in the new Supplementary Note 3.

We agree that it would be interesting to learn why distributions of disordered regions in novel and previously known RBPs differ across species and agree that the possibility that this might reflect similarities between mouse embryonic stem cells and *Drosophila* embryos regarding disordered regions is interesting. As biases in annotation of disordered regions in different species, as well as biological factors, could contribute to the observed effect, we think

a rigorous comparison is currently not possible within the scope of our study. Therefore, rather than speculating on the link between similarity in disorder and special features of RBPs at embryonic stages in different organisms, we have added the following paragraph to the manuscript (page 10).

“We noted that although the whole RNA interactome is enriched in disordered, low complexity and repetitive regions compared to the total proteome, the newly discovered proteins are on average less disordered than previously known RBPs (Fig. 2d). A similar pattern was observed for the RBPs identified in mouse embryonic stem cells ⁸, while the novel human RBPs were, on the contrary, more disordered than those previously known ¹. These observations might reflect differences in RBP functions, the mechanisms that drove evolution of RBPs in these species, or annotation biases. Although our observations call for further investigation, we find it highly probable that the difference in disorder of novel and previously known RBPs in *Drosophila* is not caused by experimental bias, considering that other parameters such as average length, isoelectric point and hydrophobicity are similar for these two groups (Supplementary Fig. 2e-g). In addition, amino acid composition of the two groups of RBPs – novel and previously known – is similar, and the significant enrichment of positively charged residues observed in both groups is characteristic of RBPs (Supplementary Fig. 2h,i).”

8. Interactome comparison, pg 10, Fig. 2e. The authors state that they compared the interactomes of human, mouse, worm and yeast. However, the Venn diagram of Fig 2e only depicts overlap with human and mouse datasets. The authors should take the opportunity and provide a more profound

discussion of data comparison in light of evolutionary conservation, which will certainly be of interest to the reader.

A Venn diagram comparing interactomes of all five species for which RNA interactome data are available is now presented in Supplementary Fig. 2j. As this diagram illustrates the same finding as the Venn diagram shown in Fig 2e (similarity between closely related species, such as mouse and human), we prefer to keep this simplified version of the diagram in the main figure for easier readability and to provide the Venn diagram of all five species as a supplement (Supplementary Fig. 2j). Furthermore, Fig. 2f shows what we consider the important information contained in Supplementary Fig. 2j, namely, the proportions of proteins carrying known RBDs and associated with the GO term “RNA binding” in flies only (left pair of bars) and in flies and one or more species (as indicated in labels below the bars). A new supplementary file (Supplementary File 2), which has been added to the revised version of the manuscript, contains the *Drosophila* RNA interactome and for each protein and indicates whether it has orthologs in worm, mouse, human or yeast, and whether these orthologs are found in the corresponding interactomes. We have also expanded the Discussion by including the following text (page 19):

“Our comparison of the *Drosophila* interactome with the data available for four other species has revealed that proteins that are conserved and appear in all interactomes are predominantly previously known RBPs involved in core processes of RNA metabolism such as translation and processing, whereas species-specific proteins are enriched for novel RBPs. This also

highlights the utility of an experimental technique such as interactome capture for discovery of novel, species-specific RPBs.”

9. Localization data, pg. 11. Interesting to see the relation to neuroblast, however the current presentation of the data is not very enlightening. Is there a statistically relevant overrepresentation of localized proteins?

We have performed statistical enrichment analysis as suggested by the reviewer, however, the findings of this analysis did not provide insight into the functions of the newly discovered RBPs. We would not include these results in the manuscript as we do not find them particularly informative.

We used published mRNA spatiotemporal expression data⁹ and determined the occurrence of specific localization patterns for each mRNA encoding a *Drosophila* RNA interactome protein. We performed a pairwise t-test comparing the fraction of novel RBPs in mRNAs associated with a particular localization category such as “Yolk cortex localization” vs. the fraction of mRNAs encoding novel RBPs in all the other categories (Figure for Reviewer 1a). We found that mRNAs encoding novel embryo RBPs are statistically enriched in the category “maternal”, and in categories “degraded” and “ubiquitous unlocalized”, which heavily overlap with the category “maternal” (Figure for Reviewer 1b), and categories “yolk cortex enrichment” and “yolk cortex localization” which are also predominantly maternal. The fraction of novel RBPs was 1.8 times or higher in each of the RNA localization categories presented in the Rebuttal Fig 1a, such as “maternal”, “degraded”

and “ubiquitous”, than among mRNAs that did not belong to these categories (>10% compared to about 5%). These results are expected, since half of the analyzed sample was generated from 0–1 h embryos, which contain almost exclusively maternal molecules.

Despite our attempts, we could not find a category of localized mRNAs that would be a) statistically enriched for the novel RBPs and b) would comprise genes involved in similar or related biological processes, so that one could hypothesize in a meaningful way about functions of the genes appearing in that category. One of the reasons could be that such functional localization categories, for example, “posterior localization”, “microtubule-associated”, etc. contain too few genes, and statistical enrichment analysis does not yield significant results due to the small size of these categories. Nevertheless, we find the localization data by Lécuyer et al.⁹ and Jambor et al.¹⁰ a useful tool to build hypotheses on functions of individual proteins, as illustrated by the example of neuroblast-localized mRNAs mentioned in the manuscript.

Figure for Reviewer 1

(a) Fractions of RBP-coding mRNAs belonging or not belonging to certain mRNA categories defined by Lécuyer et al.⁹. Right column represents background distribution, and if it is significantly different (pairwise t-test) from the fraction in the left column, the category is considered statistically enriched or depleted of RBP-coding mRNAs. Only categories for which the t-test gives a significant result are represented. (b) Venn diagram showing overlap between three mRNA categories defined by Lécuyer et al. (2007). (c) Fractions of novel RPBs in the three categories statistically enriched for novel RBP – similar to (a).

10. Labeling and legend for figure 3 need revision and editing. If applicable, the authors should quantify data to get an idea what fraction of cycB or EB1 is bound to RNA in cells?

We apologize for the lack of clarity due to the over-succinct labeling of the figure and legend text. We have revised the labeling of the figure and now explain each panel in detail in the legend. If we could, we would indeed quantify the fraction of each protein (e.g. cycB and EB1) bound to RNA in the embryo. However, in view of the experimental procedure, it is unfortunately not possible to do so: quantification of the RNA-bound fraction requires knowledge of the cross-linking efficiency, which depends on the nucleotide and amino acid composition of the RNA and interacting protein, respectively, at the contact site¹¹, and which varies at different depths due to absorbance of UV light by the embryo¹².

11. The presence of many splicing factors among the group 3 proteins is intriguing and very interesting. Moreover, the observation that alternative exon often code for RBDs or disordered region. The authors should try to further consolidate this finding - for instance, they could look at their MS data for the 38 proteins - is there direct evidence for the presence of the alternative spliced forms?

It is correct that alternative exons frequently encode domains, among which are several RBDs (Fig. 5d) and, in fact, some of the 38 dynamic binders contain alternative exons that encode RBDs. However, we did not include these findings, which are based on transcriptome sequencing, as they were

not matched with the proteomic data. Specifically, in our mass-spectrometry data we did not find peptides that fulfilled the two requirements: (a) the peptide is encoded by an alternative exon, (b) the peptide's abundance change during development follows the change in inclusion rate of the alternative exon. Our inability to identify such peptides might be due to insufficient depth of the proteomic data: not all peptides that may theoretically result from a proteins' protease cleavage were detected. We have included a short comment on page 17:

“According to our transcriptomic data, some RBPs are encoded by alternatively spliced mRNAs, and alternative exons contain RBDs. However, we could not find direct evidence for alternative inclusion of RBDs by mass-spectrometry, likely due to insufficient depth of the proteomic data.”

12. Discussion pg 17. The argument that TMT MS2 allows efficient reduction of experimental noise seems a bit far reached. At the end, the semi-quantitative approach was beneficial to come-up with a list of mRBPs. TMT is suitable for comparing treated and untreated samples, however without strict further validation of the data - no judgement on the performance of this versus other approaches can be made in my opinion.

In this study we used an MS3 approach for peptide quantification¹³. The resulting MS3 spectra are recorded at high resolution/high mass accuracy, consequently they contain little background noise, which ultimately results in many missing values in the no crosslinking channels. This is not exclusive to our study but holds true for any kind of affinity purification-mass

spectrometry (AP-MS) approach that utilizes TMT MS3. Due to the amount of missing data points we chose not to impute the missing values in order to obtain ratios for all peptides, but rather selected a semi-quantitative approach. This is the key difference to MS1-based quantification approach: when it is applied AP-MS data, non-zero background noise of MS1 spectra allows calculation of enrichment ratios.

Reviewer #2 (Remarks to the Author):

In this manuscript, Sysoev et al provide the first global in vivo identification of RBPs in *Drosophila*. They achieve this using the "RNA interactome capture" method first developed by Hentze and colleagues in human tissue culture cells, which has also been used to identify RBPs in mouse, *C. elegans*, and yeast. The identification of *Drosophila* RBPs presented here complements these previous studies and provides additional insights into the conservation of RBPs in eukaryotes. Moreover, the work presented here provides an assessment of the dynamics of the RNA-binding proteome during a developmental transition - the maternal-to-zygotic transition during *Drosophila* embryogenesis (although see caveats listed below).

The experiments and analysis appear generally sound (again, see caveats listed below), and the list of *Drosophila* RBPs obtained represents an important resource. However, various technical aspects of the study require further clarification, additional analyses should be performed to more thoroughly assess the properties of the identified RBPs, and several caveats

need to be included when presenting the data and conclusions, as described below:

1) 254nm UV irradiation penetrates only a short distance into *Drosophila* embryonic cytoplasm (see for example, Togashi and Okada 1983 *Dev Growth Diff*). Are the authors sure that their UV treatment crosslinked RBPs to RNAs throughout the embryo, as opposed to mostly near the periphery? If not this caveat must be mentioned.

Indeed, it has been determined that UV light penetrates only to a depth of approximately 5 μm in the embryo¹², where its intensity is reduced to 10% of its surface intensity. We therefore presume that our data is biased towards proteins located near the embryo surface and we mention this caveat in the main text (page 5):

“Although 254 nm UV light does not reach the deepest volume of the embryo¹², it was the only agent suitable for reproducible and efficient cross-linking of RNA to protein in live embryos.”

UV light is not the only available cross-linking agent capable of stabilizing RNA-protein interactions. Formaldehyde and other chemicals can also induce formation of covalent bonds between RNA and proteins, however their use for fixing interactions in living embryos is limited, if not impossible, for at least three reasons. First, chemical agents are difficult to deliver into a living embryo: such a procedure involves dechorionation of embryos, permeabilization of the viteline membrane, and prolonged incubations in a solution containing the crosslinker. In addition to not guaranteeing successful

and reproducible cross-linking, such treatment would inflict stress upon embryos and cause artifacts. Second, formaldehyde, as well as other chemicals, causes substantial degradation when doses required for capture of sufficient amounts of proteins are applied. Use of formaldehyde on embryo lysates might yield more reproducible results, however, the relevance of captured interactions to the native state would be questionable. Third, chemical agents cause protein-protein cross-linking at a detectable rate. We therefore considered UV light an optimal agent for cross-linking of RNA to direct protein binders in vivo, despite the limitations discussed below. Finally and most importantly, the use of UV light as a cross-linker allows us to compare our dataset with the other existing RNA interactomes, which were generated using UV crosslinking-based approach.

It is possible that some of the known RPBs expressed in the early embryo evaded capture because they are localized deep in the middle. This is one of the limitations of our method that should not, however, shadow its advantages. A brief discussion of this issue can be found in the revised manuscript (page 19):

“Some well-known RBPs expressed in the early embryo were not captured in our experiments, possibly because of their association with deadenylated mRNAs that are present at early stages¹⁴, or the limited ability of the 254 nm UV light to penetrate the optically dense embryo and, therefore, activate nucleobases of RNAs located in its deepest layers¹².”

2) The need to rely on the polyA tail for RBP capture limits the ability to capture all RBPs at any individual stage, and in particular is problematic for the purposes of comparing the RNA-bound proteome at early and late timepoints during the maternal-to-zygotic transition. Changes in tail length are a major feature of this transition with regard to the regulation of mRNA decay and translational control. Oligo-dT selection will preferentially purify mRNAs with long (rather than short) polyA tails. Thus, the population of mRNAs purified and therefore the associated RBPs is biased. Furthermore, if RBPs are involved in regulating tail length, and bind to targets with different length tails at the early versus late timepoints, this may appear artificially as a change in the RBP's binding activity. This possibility is particularly relevant with regard to the class 3 "dynamic binders" identified by the authors. The authors need to include discussion of these caveats both in the Results and Discussion.

We agree that our method does not capture proteins bound to deadenylated RNAs, and we discuss this in the context of interactome determination on page 19 (see comment 1, above) and in the context of comparison of the RNA-bound proteomes at two stages on page 21.

“Possible active regulation mechanisms include, but are not limited to allosteric regulation by trans-acting factors such as proteins, RNAs or small molecules, competitive binding of trans-acting factors at the RNA-binding site and post-translational modifications. Some RBPs might also appear to bind RNA more efficiently at one of the stages if the mRNAs that they bind have undergone a change in polyadenylation status.”

3) I am confused as to how the authors used the different timepoints in their initial definition of the RNA interactome. My understanding is that they were treated as pooled samples at this stage to define the interactome, and separate samples later on for assessing the dynamics of RNA binding. The authors should explain these details more clearly.

It is correct that for the initial identification of the RNA interactome, early and late samples were pooled, and subsequently the same samples were treated separately to identify changes in the RNA-bound proteome. We apologize for the confusion and have revised the description of our experimental design (page 7):

“For total interactome determination we pooled early and late samples to obtain three biological replicates – three CL and three noCL, control samples – that were analyzed in parallel using TMT MS with six different isobaric labels (TMT MS3 16, Online Methods and Supplementary Note 2).”

4) GO term enrichments among the entire interactome, as well as just the set of novel RBPs, would be informative and should be assessed. For instance, do the authors see an enrichment for metabolic enzymes as has been observed in the RNA-binding proteome of other species? Are there additional enriched functions?

We have added the GO term enrichment analysis of the whole interactome compared to the total proteome to Supplementary Fig. 2a and mentioned it in the manuscript on page 8, (translation, RNP structure organization).

We also looked for GO terms specific to the novel RBPs, as opposed to previously known ones. Expectedly, this analysis revealed that RNA-related terms such as “RNA binding”, “Splicing”, “Translation”, “RNA metabolism”, etc. were depleted among novel proteins, however, we could not find RNA-unrelated GO terms that were depleted or enriched among novel RBPs when a reasonable statistical threshold was set. There are at least two explanations for the lack of enrichment of particular GO terms among novel RBPs: a) the sizes of compared protein lists could be too small to identify statistically significant differences or b) RNA binding is a widespread feature that could be found in proteins involved in many different biological processes in diverse cellular components and harboring diverse molecular functions.

Regarding metabolic enzymes, we quote here our response to Reviewer 1, who asked a similar question:

“Indeed, our *Drosophila* RNA interactome includes many enzymes, including several metabolic enzymes. A total of 47 of the 268 proteins annotated as enzymes (Expasy enzyme database, 13-April-2016 release; <http://enzyme.expasy.org/>) were recovered as high-confidence RBPs in our experiments and are thus included in our RNA interactome. In the revised manuscript, we have added a Supplementary File 3 that lists all *Drosophila* enzymes (Expasy), indicating (a) whether they are part of the RNA interactome, (b) whether they have orthologs in mammals, yeast, or worm and, if so, (c) whether the orthologs are part of the RNA interactome of the respective organism. We also include a brief description and a discussion of

these findings in the Results and Discussion sections of the revised manuscript.

Results (page 12): “Discovery of several metabolic enzymes as RBPs in *Drosophila* is consistent with findings of other interactome capture studies ¹⁻³ and is noteworthy as energy production was shown to be an important factor in developmental processes in some organisms ^{4,5}, and its regulation might play a role in the MZT in *Drosophila* ⁶. As for the mammalian and yeast interactomes ², our *Drosophila* interactome contains a substantial number (47) of proteins listed as enzymes in the ExPASy database (<http://enzyme.expasy.org/>). Whereas in yeast all steps of the glycolytic pathway can be catalyzed by enzymes identified as RNA binders ², in *Drosophila* only two glycolytic enzymes – phosphofructokinase and phosphoglyceromutase – are found in the RNA interactome. Further experiments on individual proteins will elucidate the role(s) of RNA binding by metabolic enzymes in development.”

Discussion (page 20): “In the light of evidence indicating that regulation of metabolism might be tightly bound to the MZT in *Drosophila*,⁶ and previous reports of RNA binding by metabolic enzymes in other organisms ¹⁻³ the discovery of such enzymes in *Drosophila* is particularly interesting. It was previously shown that 37 mRNAs encoding metabolic enzymes are bound and possibly downregulated by a key MZT driver, Smaug ⁶. RNA binding by metabolic enzymes indicates that an additional post-translational mechanism might regulate their activity. The number of metabolic enzymes capable of binding RNA is lower in *Drosophila* than in other organisms for which RNA

interactomes are available. Why this might be, as well as the roles of metabolic enzymes in RNA metabolism and regulation, should be clarified by future studies.”

5) What GO terms were considered as "RNA-related"? This should be indicated somewhere.

RNA-related GO terms are all GO terms containing the word “RNA” , such as “RNA splicing”, “RNA regulation”, “RNA metabolism” (but not “RNA-binding”). We now explain this in the revised manuscript (page 8):

“134 (out of 523; 26%) RNA interactome proteins were not annotated by the GO term “RNA binding” but were nevertheless annotated with other RNA-related GO terms (any GO term that contains “RNA”, e.g. “RNA splicing”)”

6) How do novel versus previously characterized RBPs breakdown in terms of differential expression/RNA-binding at the early and late timepoints?

We have added a bar plot showing numbers of novel and previously known RBPs in each of the three dynamic classes to Supplementary Fig. 4e and mentioned it in the text on page 16. Only high confidence RBPs were considered, i.e. proteins that are part of the interactome.

Page 16: “Interestingly, classes 2 and 3 were enriched in previously known RBPs as compared to class 1, which contains the majority of the novel RBPs (Supplementary Fig. 4e).”

7) Do any of the class 3 RBPs (putative "dynamic binders" - see above) show differential inclusion of RBDs at the early and late timepoints as a result of alternative splicing, based on the sequencing analysis? If not, there is only weak evidence for the authors' assertion that differential splicing is a major contributor to the observed "dynamicity" of RNA binding. Other potential mechanisms should be discussed in more detail, (including the potential biases introduced by the use of polyA-based capture, as discussed above).

We apologize if we conveyed the impression that we consider differential splicing to be a major contributor to RNA binding dynamicity based on our results. We quote here our answer to a question on this topic raised by Reviewer 1:

It is correct that alternative exons frequently encode domains, among which are several RBDs (Fig. 5d) and, in fact, some of the 38 dynamic binders contain alternative exons that encode RBDs. However, we did not include these findings, which are based on transcriptome sequencing, as they were not matched with the proteomic data. Specifically, in our mass-spectrometry data we did not find peptides that fulfilled the two requirements: (a) the peptide is encoded by an alternative exon, (b) the peptide's abundance change during development follows the change in inclusion rate of the alternative exon. Our inability to identify such peptides might be due to insufficient depth of the proteomic data: not all peptides that may theoretically result from a proteins' protease cleavage were detected. We have included a short comment on page 17:

“According to our transcriptomic data, some RBPs are encoded by alternatively spliced mRNAs, and alternative exons contain RBDs. However, we could not find direct evidence for alternative inclusion of RBDs by mass-spectrometry, likely due to insufficient depth of the proteomic data.”

Minor comments:

Figure 1c: x-axis labels should be explained more clearly in legend.

We have added information regarding both the x-axis and the y-axis to the legend of Fig. 1c:

“On the x-axis are indicated the samples in which the amounts of the different RNAs were measured: Input noCL, Input CL, Eluate noCL, Eluate CL. The y-axis represents the fold enrichment of RNA amounts in the different samples. RNA amounts in noCL input were defined as 1.”

What does the asterisk in Figure 1d indicate?

The asterisk shows the position of the weak tubulin signal. The band is detected in the sample that was not treated with 12.5 mM DTT at 60°C and was removed completely upon treatment. The panel now appears as Supplementary Fig. 1h.

Are the first six lanes in Figure 1d and 1e serial dilutions? This should be explained somewhere.

Supplementary Figure S1, panel (a): again, are the input lanes serial dilutions at the two different UV treatments? This should be indicated.

We have revised the figures, indicating the dilutions either with a symbol above the image (e.g. Fig. 1d, Fig. 4b), or specifying the dilutions directly above the blot (Fig. 1e) or in the corresponding legends.

Figure 1h: what does "B-H" stand for?

B-H stands for "Benjamini-Hochberg". According to criteria of Benjamini and Hochberg ¹⁵, 65 proteins were significantly enriched and are therefore considered to be true RBPs, while the remaining proteins were considered high confidence hits by the semi-quantitative scoring scheme described in the main text, Online Methods and Supplementary Note 2.

Figure 2g: are these differences statistically significant?

The Fischer test p-values have been added to the figure legend. At the p-value cut-off of 0.01, the differences in proportions between the interactome and the proteome were statistically significant for proteins with lethal and sterile phenotypes: lethal – 0.00346; embryonic – 0.03483; sterile – 0.0001754.

Figure 4g: typo in graph label - "interactomle"

Thank you for pointing out this error, which we have corrected.

Figure 5b: why is there no median indicated in the box plot for "all genes"? Also, the P-value cut-off and statistical test used for the comparison should be indicated.

On the box plot in Fig. 5b, the median mark of all genes overlaps with the x-axis, therefore it is not visible.

The Student t-test was used to assess the differences between expression of all genes and targets of splicing factors. We have added the p-values to the figure legend. At the p-value cut-off of 0.01, the differences between datasets are insignificant.

Reviewer #3

1) Some gels/graphics are not well labeled. For example, in Figs 1d-e and 3a do the several lanes for "input" or "unbound" represent decreasing amounts of material? In Fig 3a, please indicate the positions of the fusion proteins alluded to in the text. In Fig 2c, please label the X axis.

We have labeled the x-axes in Fig. 2c and revised the labeling of other panels.

Fig. 3a shows immunoprecipitates of only one protein – free GFP. We apologize for giving the wrong impression that Fig. 3a contains immunoprecipitates of other GFP fusion proteins. For greater clarity we have modified the corresponding paragraph in the main text, page 13:

“Upon immunoprecipitation and extensive washing, we obtained high yields of GFP proteins, as illustrated by the case of free GFP in Fig. 3a. Following partial digestion, cross-linked RNA was ³²P labeled using T4 polynucleotide kinase (PNK), the protein-RNA complexes separated on an SDS polyacrylamide gel and transferred onto a PVDF membrane.”

2) In Fig 3d the n^o lanes in the autoradiography and the Western blot is not the same. Are these the same gels?

The two images are of the same membrane (same gel). In fact, the labeling of the figure was incorrect and we have revised it. We apologize for the confusion.

3) The authors use 1 J/cm² as the energy of crosslinking, but Fig S1a shows a better crosslinking efficiency at 2 J/cm². Did the authors find RNA degradation at this energy?

Although treatment 2 J/cm² slightly increases RBP yields compared to samples treated with 1 J/cm², and no RNA degradation is observed under either of the two cross-linking conditions, we chose 1 J/cm² to reduce sample processing times. A detailed discussion of the issue can be found above (Reviewer 1, point 1).

4) Page 10, last lane: I guess the authors mean Figure 1f (not 1b)

We have checked and corrected the figure references throughout the text.

5) Page 8: The numbers for the total embryo proteome do not coincide with those shown in Fig 2a.

Here we quote our response to a similar comment by Reviewer 1 (comment 5):

“The numbers presented as “total embryo” in the original Fig. 2a represent the total proteome minus the interactome proteins, and therefore do not add up to the total of 3655 proteins identified in the total proteome. We have revised the labeling in the revised Fig. 2a, such that this is clearly spelled out. It is important to note here that not all interactome proteins were identified in the total proteome. We have also added a scale bar indicating the fraction of proteins (left of panel). We apologize for the confusion.”

6) Page 42, first paragraph: Please, explain better the FDR threshold that was selected to consider proteins not changing in abundance.

We have revised the corresponding paragraph in the Supplementary Note 2, adding information:

“To identify dynamic RNA-binding proteins, the quantitative analysis of the differential binding was compared to the differential total proteome. The differentially binding proteins at FDR 10% were separated in two classes, those that change binding, because of a change in protein abundance, and those that change in binding but do not change protein abundance. The latter we called “dynamic binders”. To identify a set of proteins not changing in abundance, we selected all proteins above an FDR threshold of 43.6%. For a

set of proteins one can estimate the absolute number of changing proteins by subtracting the expected false discoveries from the size of the set. The threshold is chosen such that the absolute number of changing proteins is maximized.”

References

1. Castello, A. *et al.* Insights into RNA Biology from an Atlas of Mammalian mRNA-Binding Proteins. *Cell* **149**, 1393–1406 (2012).
2. Beckmann, B. M. *et al.* The RNA-binding proteomes from yeast to man harbour conserved enigmRBPs. *Nat Commun* **6**, 10127 (2015).
3. Matia-González, A. M., Laing, E. E. & Gerber, A. P. Conserved mRNA-binding proteomes in eukaryotic organisms. *Nat. Struct. Mol. Biol.* (2015). doi:10.1038/nsmb.3128
4. Harvey, A. J., Kind, K. L. & Thompson, J. G. REDOX regulation of early embryo development. 1–8 (2002).
5. Snaebjornsson, M. *et al.* A role for central carbon metabolism in mammalian embryonic development? in 1–1 (2014). doi:10.1186/2049-3002-2-S1-P69
6. Chen, L. *et al.* Global regulation of mRNA translation and stability in the early *Drosophila* embryo by the Smaug RNA-binding protein. 1–21 (2014).
7. Vizcaíno, J. A. *et al.* 2016 update of the PRIDE database and its related tools. *Nucleic Acids Research* **44**, D447–56 (2016).
8. Kwon, S. C. *et al.* The RNA-binding protein repertoire of embryonic stem cells. 1–11 (2013). doi:10.1038/nsmb.2638
9. Lécuyer, E. *et al.* Global analysis of mRNA localization reveals a prominent role in organizing cellular architecture and function. *Cell* **131**, 174–187 (2007).
10. Jambor, H. *et al.* Systematic imaging reveals features and changing localization of mRNAs in *Drosophila* development. *Elife* **4**, (2015).
11. Saito, I. & Matsuura, T. Chemical aspects of UV-induced crosslinking of proteins to nucleic acids. Photoreactions with lysine and tryptophan. *Acc. Chem. Res.* **18**, 134–141 (1985).
12. Togashi, S. & Okada, M. Effects of UV- irradiation at Various Wavelengths on Sterilizing *Drosophila* Embryos. *Development, Growth & Differentiation* **25**, 133–141 (1983).
13. Ting, L., Rad, R., Gygi, S. P. & Haas, W. MS3 eliminates ratio distortion in isobaric multiplexed quantitative proteomics. *Nat. Methods* **8**, 937–940 (2011).
14. Sallés, F. J., Lieberfarb, M. E., Wreden, C., Gergen, J. P. & Strickland, S. Coordinate initiation of *Drosophila* development by regulated polyadenylation of maternal messenger RNAs. *Science* **266**, 1996–1999 (1994).
15. Benjamini, Y. & Hochberg, Y. Controlling the False Discovery Rate: A

Practical and Powerful Approach to Multiple Testing. *Journal of the Royal Statistical Society* **57**, 289–300 (1995).

Reviewers' Comments:

Reviewer #1 (Remarks to the Author)

The authors have addressed the referee's comments in great detail. Figures have been updated, texts revised and proteomics datasets deposited for access to the public.

There are two final minor points that should be considered prior publication.

1. It is appreciate that the authors discuss metabolic enzymes identified in their screen. However, the new text needs occasionally more complete references - especially the more recent work.

On page 12, the authors write:

"As for the mammalian and yeast interactomes 10, our *Drosophila* interactome contains a substantial number (47) of proteins..."

..."Whereas in yeast all steps of the glycolytic pathway can be catalyzed by enzymes identified as RNA binders 10, in *Drosophila* only two glycolytic enzymes..."

In both cases, the authors referenced only their own work Ref. 10 (Beckman et al) excluding others. For instance, reference [11](Matia-Gonzalez et al. 2015) showed that all yeast glycolytic enzymes bind to mRNAs, with some of them being glycolytic mRNAs. Therefore, Ref 11 must be added to provide the reader with appropriate references.

2. When this publication was under revision, a likewise study has been published by the Landthaler laboratory describing the mRNA interactome in *Drosophila* embryos (Wessels H.H. et al. *Genome Res.* Published in Advance April 28, 2016, doi:10.1101/gr.200386.115). The authors should reference this new work. Although it is not be expected that the authors discuss their work in light of this related work in detail, it would be good to add a few sentences briefly outlining common and different findings between the studies (e.g. for instance describe the overlap of identified mRPBs).

Reviewer #2 (Remarks to the Author)

The authors have responded satisfactorily to my concerns and have made appropriate revisions. The manuscript should be accepted for publication.

Reviewer #3 (Remarks to the Author)

The manuscript by the Ephrussi and Hentze labs has improved with the new revisions. The authors have satisfactorily addressed the reviewers' comments. This work represents the first comparative interactome capture study, and the first resource of RBPs in early *Drosophila* development. As such, the study should be useful for the RNA and Development communities at large.

We thank the reviewers for their insightful and constructive comments throughout the revision process.

Reviewer #1 (Remarks to the Author):

1. “Authors referenced only their own work Ref. 10 (Beckman et al) excluding others. For instance, reference [11](Matia-Gonzalez et al. 2015) showed that all yeast glycolytic enzymes bind to mRNAs, with some of them being glycolytic mRNAs. Therefore, Ref 11 must be added to provide the reader with appropriate references.”

We are grateful to the reviewer for pointing out this unintentional omission and have added the missing reference according to his/her suggestion.

2. When this publication was under revision, a likewise study has been published by the Lanthaler laboratory describing the mRNA interactome in Drosophila embryos (Wessels H.H. et al. Genome Res. Published in Advance April 28, 2016, doi:10.1101/gr.200386.115). The authors should reference this new work. Although it is not be expected that the authors discuss their work in light of this related work in detail, it would be good to add a few sentences briefly outlining common and different findings between the studies (e.g. for instance describe the overlap of identified mRPBs).

Indeed, a new publication from the Lanthaler lab has appeared online as an “Accepted manuscript”, in manuscript format. As supplementary information is not yet available for this publication, we could only perform a preliminary comparison between the data described by Wessels et al. and our dataset. In the revised version of our manuscript we have added a paragraph referring to the Wessels et al. study.